



# Technical Note: Evaluation of the Copernicus Atmosphere Monitoring Service Cy48R1 upgrade of June 2023

Henk Eskes[1], Athanasios Tsikerdekis[1], Melanie Ades[2], Mihai Alexe[2], Anna Benedictow[3],
Yasmine Bennouna[4], Lewis Blake[3], Idir Bouarar[5], Simon Chabrillat[6], Richard Engelen[2],
Quentin Errera[6], Johannes Flemming[2], Sebastien Garrigues[2], Jan Griesfeller[3], Vincent Huijnen[1],
Luka Ilić[7], Antje Inness[2], John Kapsomenakis[8], Zak Kipling[2], Bavo Langerock[6], Augustin Mortier[3],
Mark Parrington[2], Isabelle Pison[9], Mikko R.A. Pitkänen[10], Samuel Remy[11], Andreas Richter[12],
Anja Schoenhardt[12], Michael Schulz[3], Valerie Thouret[4], Thorsten Warneke[13], Christos Zerefos[8], and
Vincent-Henri Peuch[2]

[1]Royal Netherlands Meteorological Institute, De Bilt, the Netherlands
[2]European Centre for Medium-Range Weather Forecasts, Reading, UK
[3]Norwegian Meteorological Institute, Oslo, Norway
[4]CNRS and Université Paul Sabatier, Laboratoire d'Aérologie, Toulouse, France
[5]Max Planck Institute for Meteorology, Hamburg, Germany
[6]Belgian Institute for Space Aeronomy, BIRA-IASB, Brussels, Belgium
[7]Barcelona Supercomputing Centre, Barcelona, Spain
[8]Academy of Athens, Athens, Greece
[9]Laboratoire des Sciences du Climat et de l'Environnement, Paris, France
[10]Finnish Meteorological Institute, Helsinki, Finland
[11]HYGEOS, Lille, France
[12]Institute of Environmental Physics, University of Bremen, Bremen, Germany
[13]University of Bremen Campus GmbH, Bremen, Germany

**Correspondence:** Henk Eskes (henk.eskes@knmi.nl)

**Abstract.** The Copernicus Atmosphere Monitoring Service (CAMS) is providing daily analyses and forecasts of the composition of the atmosphere, including the reactive gases such as $O_3$, CO, $NO_2$, HCHO, $SO_2$, aerosol species and greenhouse gases. The global CAMS analysis system (IFS-COMPO) is based on the ECMWF Integrated Forecast System (IFS) for numerical weather prediction (NWP), and assimilates a large number of composition satellite products on top of the meteorological
5   observations ingested in IFS. The CAMS system receives regular upgrades, following the upgrades of IFS. The last upgrade, Cy48R1, operational since 27 June 2023, was very major with a large number of code changes, both for COMPO and for NWP. The main COMPO innovations include the introduction of full stratospheric chemistry, a major update of the emissions, of the aerosol model, including the representation of secondary organic aerosol, several updates of the dust life cycle and optics, inorganic chemistry in the troposphere, and the assimilation of VIIRS AOD and TROPOMI CO. The CAMS Cy48R1
10  upgrade was validated using a large number of independent measurement datasets, including surface in situ, surface remote sensing, routine aircraft and balloon and satellite observations. In this paper we present the validation results for Cy48R1 by comparing with the skill of the previous operational system (Cy47R3), with the independent observations as reference, for the period October 2022 to June 2023 during which daily forecasts from both cycles are available. Major improvements in



skill are found for the ozone profile in the lower-middle stratosphere and for stratospheric $NO_2$ due to the inclusion of full
stratospheric chemistry. Stratospheric trace gases compare well with ACE-FTS observations between 10-200 hPa, with larger
deviations between 1-10 hPa. The impact of the updated emissions is especially visible over East Asia and is beneficial for the
trace gases $O_3$, $NO_2$, and $SO_2$. The CO column assimilation is now anchored by IASI instead of MOPITT which is beneficial
for most of the CO comparisons, and the assimilation of TROPOMI CO data improves the model CO field in the troposphere.
In general the aerosol optical depth has improved globally, but the dust evaluation shows more mixed results. The results of the
47 comparisons are summarised in a score card, which shows that 83% of the evaluation datasets show a neutral or improved
performance of Cy48R1 compared to the previous operational CAMS system, while 17% indicate a (slight) degradation. This
demonstrates the overall success of this upgrade.

## 1 Introduction

The Copernicus Atmosphere Monitoring Service (CAMS, http://atmosphere.copernicus.eu) is the service component of the
European Earth Observation programme Copernicus focussing on atmospheric composition (Peuch et al., 2022). The CAMS
system was developed during a sequence of European research projects (Hollingsworth et al., 2008) and became an operational
service in 2015. The CAMS global near-real time (NRT) service provides daily analyses and 5-day forecasts of reactive trace
gases and aerosol concentrations, and delayed-mode greenhouse gas analyses and forecasts. CAMS makes use of the mea-
surements of the fleet of Copernicus Earth observation satellites, the Sentinels, as well as other relevant satellite and surface
observations. Apart from the near-real time products CAMS produces global reanalyses for reactive gases and aerosols (Inness
et al., 2019; Flemming et al., 2017) and greenhouse gases (Agustí-Panareda et al., 2023).

The global CAMS system is part of the Integrated Forecasting System (IFS) of ECMWF, the system used to produce the
medium-range Numerical Weather Predictions (NWP). The modelling of reactive gases, aerosols and greenhouse gases is fully
integrated in IFS (ECMWF, 2023). The assimilation of composition satellite data is part of the IFS 4D-Var data assimilation
suite, and CAMS is assimilating the complete set of meteorological observations together with the composition data. CAMS
forecasts are therefore combined weather-composition forecasts. The extended IFS system developed by CAMS for trace gases
and aerosols is referred to as IFS-COMPO. A second CAMS system, IFS-GHG, is developed for $CO_2$ and $CH_4$ analyses,
(high-resolution) forecasts, and reanalyses. This paper focuses on the evaluation of the IFS-COMPO NRT products.

The CAMS effort (Peuch et al., 2022) includes dedicated scientific activities to continuously develop and improve the
modelling and satellite assimilation aspects of IFS-COMPO, including the chemistry code (Flemming et al., 2015; Huijnen
et al., 2016, 2019), and the development of the aerosol scheme (Rémy et al., 2019, 2022). The data assimilation of reactive gases
is described in Inness et al. (2015). Recent extensions relevant for Cy48R1 include the modelling of stratospheric chemistry
(Huijnen et al., 2016; Chabrillat et al., 2023), the assimilation of TROPOMI CO (Inness et al., 2022b) and VIIRS AOD
(Garrigues et al., 2023). Since Cy48R1 the IFS upgrade documentation also includes a chapter with a detailed discussion of
the IFS-COMPO related modelling and data assimilation changes (ECMWF, 2023).



The CAMS service makes extensive use of independent measurement datasets of proven quality, available for different parts of the world, to evaluate the quality of the forecast/analysis and reanalysis products (Eskes et al., 2015; Wagner et al., 2021; Lefever et al., 2015; Cuevas et al., 2015; Agustí-Panareda et al., 2023). In total of the order of 65 measurement collections are used, obtained from observational networks like NDACC, WMO-GAW, AERONET, IAGOS, ICOS, IASOA, and space organisations like NASA, ESA, EUMETSAT and others. The CAMS global real-time service component is evaluated in a series of validation reports produced every 3 months (Benedictow et al., 2023). The CAMS reanalysis validation reports for the aerosols and reactive gases (Bennouna et al., 2023) and greenhouse gases (Ramonet et al., 2021) are updated on a yearly basis. New system upgrades are evaluated before becoming operational, where CAMS Cy47R3 is discussed in Eskes et al. (2021), and Cy48R1 in Eskes et al. (2023b).

The ECMWF NWP and CAMS operational systems are upgraded at the same time and follow the same model cycles. A CAMS upgrade implies therefore a simultaneous upgrade of the NWP part of IFS in IFS-COMPO and IFS-GHG. The last upgrade, and topic of this paper, is the upgrade from Cy47R3 to Cy48R1, implemented on 27 June 2023. Upgrades normally occur at least once a year, but Cy48R1 is packing almost two years of developments. The delay is linked to the move of the ECMWF high-performance computing facilities from Reading, UK, to Bologna, Italy. As a result, Cy48R1 represents a very major upgrade both for NWP and for CAMS.

Before each upgrade, the new analysis and forecast configuration is operated as a so-called experimental suite or e-suite in parallel to the operational near-real time service (the o-suite or compo-suite, e.g., Cy47R3 in this paper) for about half a year. For the 27 June 2023 upgrade to Cy48R1, the e-suite run is available from 1 October 2022 to the moment of the upgrade, 27 June 2023. This implies that there are only a limited number of 9 months of data in the Autumn, Winter and Spring available to the evaluation, with a focus on wintertime. CAMS is also producing control runs without the assimilation of atmospheric composition data. The control run allows us to distinguish changes due to the model from changes due to the assimilation of the atmospheric composition observations.

In this paper, the validation results are presented from comparisons of the performance of the Cy48R1 e-suite runs and e-suite control run with the operational run (o-suite and o-suite control run), all compared with independent observations. Prior to the upgrade, a preliminary evaluation of the new cycle was presented in a CAMS report (Eskes et al., 2023b), but this evaluation covered a relatively short period of about 5 months of CAMS results that were available at the time. This paper extends the validation by several months by including e-suite and o-suite analyses and forecasts up to the day of the upgrade, leading to more refined conclusions.

In section 2 we summarise the changes implemented in Cy48R1, and in section 3 we provide an overview of the validation datasets used. The validation results are shown in section 4 in the form of a series of plots comparing e-suite, o-suite and independent observations.





## 2 Overview of the CAMS global system

The operational CAMS global system provides analyses of the atmospheric composition of aerosols and reactive gases worldwide by blending satellite data and atmospheric simulation through a process called data assimilation. An overview of all
satellite datasets assimilated in the CAMS global system can be found in Peuch et al. (2022) and Eskes et al. (2023b). Further details on the different production runs and their data usage can be found at the CAMS global products website (ECMWF, 2023e). The history of CAMS system upgrades, the data products, the satellite data assimilated (and monitored) and relevant references, is available in the CAMS data documentation website (ECMWF, 2023b), while the results of the operational satellite data monitoring is available at the CAMS data monitoring website (ECMWF, 2023c).

Copernicus products are made available for free. The CAMS products can be accessed through the Atmosphere Data Store (ADS) at https://ads.atmosphere.copernicus.eu/. The available o-suite output includes: analysis fields available every 3 hours, for the two 12-hour analysis windows per day; twice daily 5-day forecast starting from the analysis at 00 and 12 UTC, with 2D fields available hourly, and 3D fields available every 3 hours. The control run archive contains 5-day forecasts starting from the 00 UTC analysis. The output includes a large number of trace gases, aerosol composition and optical properties of the
aerosols, as detailed on the ADS. Most of the comparisons documented below make use of the first 24 hours of the forecast runs starting 00 UTC, which allows a direct comparison with the control run. For some comparisons the analysis results were used, as indicated.

The global forecasting system is continually being evaluated to ensure the output meets the expected requirements. Comprehensive Evaluation and Quality Assurance (EQA) reports are provided for the CAMS system on a quarterly basis (ECMWF,
2023d). CAMS uses a multitude of independent data sets to routinely monitor its global forecasts. It works with various data providers, acquiring the observations with appropriate timelines and generating graphics that show the differences between the forecasts and the independent observations (ECMWF, 2023g). The two main websites are the global evaluation server for near-real time analysis and forecast products (ECMWF, 2023f) and the AeroVal evaluation server to evaluate the reanalysis products (ECMWF, 2023a).

A control run is produced both for the e-suite (e-suite control run) and o-suite (o-suite control run). These runs are useful to distinguish the impact of data assimilation changes from the impact of modelling changes. The control run applies the same settings as the respective o-suite and e-suite, but the data assimilation is not switched on for atmospheric composition. The meteorology in the control run is initialised with the meteorological fields from the o-suite (or e-suite).

In addition, the pre-operational analyses and forecasts of $CO_2$ and $CH_4$ use an independent setup of the IFS. The upgrade
of the greenhouse gas system to Cy48R1 is foreseen to take place in fall 2023, and the evaluation only just started when this paper was written. Therefore the greenhouse gas products will not be discussed in this paper.

In the next subsections the model, data assimilation and emissions upgrades of the CAMS e-suite (Cy48R1) are summarised, and the observations used are introduced. The CAMS Cy48R1, called "e-suite" in this paper, succeeded the CAMS o-suite (Cy47R3) on 27th of June 2023 as the operational version for the CAMS forecast and data assimilation system.





## 2.1 The CAMS e-suite (Cy48R1)

This upgrade encompasses several significant scientific advancements. A detailed documentation of the CAMS Cy48R1 upgrade for aerosols and reactive gases is available at ECMWF (2023). More information regarding the aerosol/chemistry changes and the meteorological changes are provided in IFS (2023) and CAMS (2023) respectively. In the following subsections the model, assimilation and emissions updates are summarised.

### 2.1.1 Model updates

The CAMS IFS cycle Cy48R1 is based on the ECMWF's IFS cycle Cy48R1. The model consists of gas-phase chemistry modules for the troposphere (based on CB05 scheme, see Williams et al. (2022) and references therein) and the stratosphere (based on the BASCOE scheme, see Errera et al. (2019), Huijnen et al. (2016) and references therein) that includes 123 active tracers. The distinction between the tropospheric and stratospheric chemistry schemes occurs at the tropopause, which is determined based on the temperature lapse rate of the model. The aerosol scheme AER (originally based on LOA/LMDZ model, Reddy (2005)) is a bulk-bin scheme that consists of 16 active species, which are coupled with the chemistry schemes in various ways (ECMWF, 2023).

Before Cy48R1 only ozone was modelled in the stratosphere using a linear parameterization. The stratospheric chemistry module activated in Cy48R1 (Huijnen et al., 2016) is a re-implementation of the Belgian Assimilation System for Chemical ObsErvations (BASCOE) chemical module and it involves 64 species engaging in 157 gas-phase, 9 heterogeneous, and 53 photolytic reactions (Errera et al., 2019). It encompasses ozone-depleting substances, greenhouse gases, and other species vital for comprehensive stratospheric ozone photochemistry (Chabrillat et al., 2023). Additionally, basic sulphur chemistry is included to represent gas-phase sulphuric acid formation and enable coupling with the sulphate aerosol module, featuring OCS, $SO_2$, $SO_3$, and $H_2SO_4$ reactions.

In the troposphere, inorganic chemistry was updated, hydrogen cyanide (HCN) and acetonitrile ($CH_3CN$) are now included as long-lived tracers, serving as indicators of biomass burning activity. For the degradation of organic compounds, the basic isoprene oxidation scheme was replaced with a more explicit approach based on Stavrakou et al. (2010), further modified according to Lamarque et al. (2012) and Myriokefalitakis et al. (2020). This updated scheme contains reaction products including glyoxal (CHOCHO), glycolaldehyde, isoprene-peroxide, hydroxy-acetone, and two hydroxy-aldehydes (Williams et al., 2022). The scheme was further expanded to include an explicit parameterization of aromatics tracers xylene and toluene, acting as precursors for Secondary Organic Aerosol (SOA).

Recent developments of the CAMS aerosol modelling are described in Rémy et al. (2022). The Cy48R1 upgrade introduced notable changes in aerosol. The SOA species are now represented with dedicated tracers, distinguishing biogenic and anthropogenic origins, and coupled with the tropospheric chemistry for their production. The e-folding time that converts hydrophobic components of Organic Matter (OM) and Black Carbon (BC) into hydrophilic forms has been decreased to 2.8 hours from 1.16 days. The assumed number size distribution for dust, which used to compute the offline dust aerosol optical properties (mass extinction, single scattering albedo and asymmetry parameter), has been updated based on values from Ryder



et al. (2018) derived from aircraft measurements over the tropical Eastern Atlantic. The refractive index has also been updated. The dust source function, that used to modulate dust emissions, has been recomputed based on a three-year analysis of the

145 MIDAS product (Gkikas et al., 2021), leading to monthly variations instead of fixed yearly values. A regional redistribution of total dust emissions into the three dust bins has been implemented, based on long simulations of dust mineralogy, which directs relatively more emissions to finer dust bins (1 and 2) compared to CY47R3. The dust mass emissions and burden are significantly higher in Cy48R1 (see details on ECMWF (2023)), which leads to an increase of dust optical depth globally by about 30%. Note however that the Mass Extinction Coefficient (MEC), which is calculated by dividing aerosol optical depth

to aerosol mass is reduced for dust in Cy48R1 (not shown). A new parameterization for the rebound effect of super-coarse dust particles over continental surfaces, relying on Zhang (2001) was added, which reduces dry deposition for those particles. In addition, sedimentation, previously limited to super-coarse dust and sea-salt, is now applied broadly to all aerosol tracers, although the impact is mainly significant in the stratosphere, where sedimentation is the dominant sink.

The aerosol optical properties of aerosols have received several updates in Cy48R1. The inclusion of a specific SOA species

distinguishes primary from secondary OM. The new set of optical properties for OM, based on Brown et al. (2018), leads to higher extinction, particularly at low relative humidity conditions. Notably, the refractive index used in Cy48R1 results in more absorbing organic matter in the UV and near-UV regions, characterised as brown carbon. Further, a scaling factor (1.375) on mass extinction of sulphate aerosol, previously based on the molar mass ratio of ammonium sulphate to sulphate, has been removed as ammonium is now a separate species since cycle 46R1.

Nitrate and Ammonium aerosols were added to the aerosol species in Cy46R1 whilst simultaneously the $SO_2$ precursor was removed. However, the update to the new species as part of the aerosol data assimilation process was not correctly included (see Benedetti et al. (2009) for a full description of the aerosol assimilation process). This was fixed in Cy48R1, leading to increments now being added to both fine and coarse nitrate and ammonium.

### 2.1.2 Assimilation updates

The assimilation of TROPOMI total column CO became operational at the $28^{th}$ of April in Cy48R1 (only the last months of the e-suite period). The impact of this inclusion was tested from July to December 2021 (Inness et al., 2022a). The results showed an 8% increase in CAMS total column CO. The assimilation impact was significant during periods of high CO emissions from boreal wildfires in July and August 2021. While TROPOMI CO assimilation enhanced column constraints, it had limited influence on individual plumes transported across continents and oceans above 500 hPa.

The CAMS aerosol data assimilation system has depended on the MODIS instrument for more than ten years. To ensure forecast continuity, as the MODIS instrument is aging, the assimilation of AOD from NOAA VIIRS AOD from S-NPP and NOAA20 was tested (Garrigues et al., 2023) and was activated in CY47R3 in February 2023 and is also active in CY48R1. Experiments assimilating VIIRS on top of MODIS or in place of MODIS in IFS cycle have been carried out from June to November 2020. A comparison with AERONET revealed that both experiments resulted in overall lower bias, notably in

Europe, Africa, and Southeast Asia, with substantial improvements over desert and maritime aerosol sites.




The assimilation of data from different satellite instruments can introduce biases compared to each other or the model. To address this, a bias correction scheme, known as Variational Bias Correction (VarBC), is employed. VarBC involves introducing additional degrees of freedom, represented as bias parameters, into the 4D-Var cost function's observational term. Observational datasets that do not use VarBC are considered anchors and are crucial for preventing drifts in the fields (Inness et al., 2013). Note that in Cy48R1 the anchor for CO was changed from MOPITT (Terra) to IASI (Metop-C) and TROPOMI (Sentinel-5p), while the new anchor for AOD is now VIIRS (NOAA-20).

Cy48R1 includes a new volcanic $SO_2$ tracer (VSO2) in addition to the $SO_2$ tracer. VSO2 is currently not coupled to the chemistry yet but uses an e-folding lifetime of 7 days. TROPOMI $SO_2$ data with layer height information are assimilated into the VSO2 tracer following the method described in Inness et al. (2022a).

On 15 December 2022, still in Cy47r3, and update of the background error covariance wavelet file was implemented to use the correct NWP background errors. The update leads to considerably improved NWP forecast scores. As a consequence, changes to AOD and upper tropospheric and lower stratospheric ozone are also expected. The comparisons presented below cover the period October 2022 - June 2023. The impact of this change is observed in ozone when comparing the first two months with the months in 2023.

### 2.1.3 Emission updates

CAMS emissions are available from gridded inventories per sector (Denier van der Gon et al., 2023) except the emissions from dust and sea salt aerosols which are calculated online (ECMWF, 2023). Most emission inventories are on a monthly resolution capturing the seasonal cycle. Only the emissions for biomass burning coming from GFAS v1.4 (Kaiser et al., 2012) are provided in daily temporal resolution, including injection heights. Specifically in Cy48R1, the model uses anthropogenic emissions from CAMS-GLOB-ANT v5.3, aviation emission from CAMS-GLOB-AIR v1.1, biogenic emissions from a climatology constructed from CAMS-GLOB-BIO v3.1 and natural emissions of DMS over the ocean from CAMS-GLOB-OCE v3.1. For reference, in Cycle 47r3 the anthropogenic emissions were based on CAMS-GLOB-ANT v4.2 and biogenic emissions from CAMS-GLOB-BIO v3.1.

Natural emissions use a monthly mean climatology. For varying volcanic $SO_2$ emissions, a climatology is constructed based on recent satellite-based inventories (Carn et al., 2017). In Cy48r1 a sector-specific treatment for any of the emissions is introduced, allowing sector-specific diurnal cycle profiles and injection heights.

## 3 Observations used for the validation of the CAMS system

The CAMS service includes activities dedicated to the validation of the global and regional (European) service products. The latest validation results for the CAMS-global near-real time service (the o-suite) products can be found in Benedictow et al. (2023) and the activity is described in Eskes et al. (2015). All CAMS validation reports for the global service products and the verification websites can be found at ECMWF (2023g).



The CAMS validation activity makes use of about 65 measurement datasets. It covers concentrations from the surface up to the stratosphere, using a wide range of instruments and measurement techniques, including surface in-situ, surface remote sensing, aircraft and balloon in-situ, and satellite observations.

The observational datasets used for the evaluation of the Cy48R1 upgrade are summarised in Table 1. A description of all these diverse measurement datasets is beyond the scope of this paper. More details on the observation networks, instruments, measurement datasets and quality control can be found in the CAMS observations document Eskes et al. (2023a) and in the list of references included in this document.

   Many of the forecast-minus-observation results show below make use of the following three metrics: the Modified Nor-
malised Mean Bias (MNMB; a symmetric and normalised form to express the mean bias), the Fractional Gross Error (FGE; a symmetric and normalised absolute mean difference), and the Correlation (R). The scoring recommendations and metrics are discussed in Tsikerdekis et al. (2023).





**Table 1.** Observational datasets used for the evaluation of the CAMS Cy48R1 e-suite.

| Instrument | Species, property | Type, region | Network, provider | URL |
|---|---|---|---|---|
| Surface in-situ | $O_3$, CO, $NO_2$, $SO_2$, PM10, PM2.5 | Europe | EEA-Airbase | https://eea.europa.eu |
| Surface in-situ | $O_3$, CO, $NO_2$, $SO_2$, PM10, PM2.5 | China | CNEMC | http://www.cnemc.cn/en/ |
| Surface in-situ | $O_3$, $NO_2$, PM10, PM2.5 | USA | AirNow | https://www.airnow.gov |
| Surface in-situ | $O_3$, CO | Global | WMO-GAW | https://community.wmo.int/en/ activity-areas/gaw |
| Surface in-situ | $O_3$ | Global | ESRL/GMD | https://www.esrl.noaa.gov |
| Surface in-situ | $O_3$ | Arctic | IASOA | https://arctic.noaa.gov/research/ international-arctic-systems-for -observing-the-atmosphere/ |
| Aircraft in-situ | $O_3$, CO | Airports | IAGOS | (http://www.iagos.org) |
| Ozone sonde | $O_3$ | Global | NDACC | https://ndacc.larc.nasa.gov |
| Surface remote sensing | $O_3$, CO | Global | NDACC | https://ndacc.larc.nasa.gov |
| Surface remote sensing | AOD, AOD coarse, AE | Global | AERONET | https://aeronet.gsfc.nasa.gov/ |
| IASI | $O_3$, CO | Satellite | EUMETSAT | https://www.eumetsat.int/iasi |
| MOPITT | CO | Satellite | NASA | https://terra.nasa.gov/ |
| TROPOMI | $NO_2$, HCHO | Satellite | ESA | https://sentinels.copernicus. eu/web/sentinel/missions/ sentinel-5p |
| ACE-FTS | Stratospheric trace gases | Satellite | CSA | http://www.ace.uwaterloo.ca |
| MLS | Stratospheric trace gases | Satellite | NASA | https://mls.jpl.nasa.gov |
| SAGE-III | O3 | Satellite | NASA | https://sage.nasa.gov |
| OMPS-LP | O3 | Satellite | NASA | https://www.earthdata.nasa. gov/sensors/omps |
| UV stations | UV-Index | Global | Collected by FMI | https://fmi.fi |





**Table 2.** Scorecard for the relative performance of the e-suite versus the performance of the o-suite against observations. Meaning of the "relative score" symbols: (++) e-suite performs significantly better than the o-suite (+) e-suite shows small improvements, (n) (neutral) no significant difference between o-suite and e-suite, (–) score is somewhat degraded in the e-suite, (––) e-suite performs significantly worse than the o-suite. Remote: Remote Sensing from surface station. [1] Based on the average statistics of ACE-FTS, SAGE-III/ISS, OMPS-LP, Aura MLS and ozonesondes. [2] Based on data from a network of stations that were collected by FMI.

| Species | Reference | Class | Type | Region | Relative Score |
|---|---|---|---|---|---|
| AOD | AERONET | Remote | Column | Global | + |
| AOD | AERONET | Remote | Column | Europe | n |
| AOD | AERONET | Remote | Column | North America | + |
| AOD | AERONET | Remote | Column | East Asia | + |
| AOD coarse | AERONET SDA | Remote | Column | Sahara | n |
| AOD coarse | AERONET SDA | Remote | Column | N. Atlantic & Mediterranean | + |
| AOD coarse | AERONET SDA | Remote | Column | Middle East | - |
| AE | AERONET | Remote | Column | Global | + |
| AE | AERONET | Remote | Column | Europe | + |
| AE | AERONET | Remote | Column | North America | + |
| AE | AERONET | Remote | Column | East Asia | + |
| AE | AERONET | Remote | Column | Sahara | n |
| AE | AERONET | Remote | Column | Middle East | - |
| PM10 | AIRBASE | In-situ | Surface | Europe | n |
| PM10 | CNEMC | In-situ | Surface | East Asia | - |
| PM10 | AIRNOW | In-situ | Surface | North America | + |
| PM2.5 | AIRBASE | In-situ | Surface | Europe | n |
| PM2.5 | CNEMC | In-situ | Surface | East Asia | - |
| PM2.5 | AIRNOW | In-situ | Surface | North America | + |
| $O_3$ | GAW, ESRL, IASOA | In-situ | Surface | Global | n |
| $O_3$ | AIRBASE | In-situ | Surface | Europe | n |
| $O_3$ | CNEMC | In-situ | Surface | East Asia | + |
| $O_3$ | AIRNOW | In-situ | Surface | North America | + |
| $O_3$ | IAGOS | Aircraft | Tropospheric Profiles | Global | n |
| $O_3$ | Sondes | Sondes | Tropospehric Profiles | Global | n |
| $O_3$ | NDACC | Remote | 1-50 hPa | Global | n |
| $O_3$ | REF[1] | Satellite | 1-10 hPa | Global | - |
| $O_3$ | REF[1] | Satellite | 10-200 hPa | Tropics | + + |
| $O_3$ | REF[1] | Satellite | 10-200 hPa | Global | + |
| $O_3$ | IASI | Satellite | Column | Global | + |
| CO | GAW | In-situ | Surface | Global | n |
| CO | AIRBASE | In-situ | Surface | Europe | + |
| CO | CNEMC | In-situ | Surface | East Asia | - |
| CO | NDACC FTIR | Remote | Tropospheric Profiles | Global | + |
| CO | NDACC FTIR | Remote | Stratospheric Profiles | Global | + |
| CO | IAGOS | Aircraft | Tropospheric Profiles | Global | + |
| CO | IASI | Satellite | Column | Global | + |
| CO | MOPITT | Satellite | Column | Global | - |
| $NO_2$ | AIRBASE | In-situ | Surface | Europe | + |
| $NO_2$ | CNEMC | In-situ | Surface | East Asia | + |
| $NO_2$ | AIRNOW | In-situ | Surface | North America | n |
| $NO_2$ | TROPOMI | Satellite | Tropospheric Column | Global | + |
| $NO_2$ | TROPOMI | Satellite | Stratospheric Column | Global | ++ |
| $SO_2$ | AIRBASE | In-situ | Surface | Europe | + |
| $SO_2$ | CNEMC | In-situ | Surface | East Asia | + |
| HCHO | TROPOMI | Satellite | Tropospheric Column | Global | - |
| UV | REF[2] | In-situ | Surface | Global | n |



# 4 Results: changes in atmospheric composition introduced by Cy48R1

In this section we will summarise the main findings of the comparison of the e-suite (Cy48R1) and o-suite (Cy47R3), where
both are evaluated with the independent observational data sets discussed above. This is presented for the individual main trace
gases and aerosol, for the available observational data sets and for regions of interest. The evaluation is done for the period
01/10/2022 to 27/06/2023 where forecast results for both cycles are available. The corresponding e-suite control run (e-control)
and o-suite control run (o-control), without the assimilation of the atmospheric composition satellite data, were also evaluated.
The improvement in performance of e-suite compared to the o-suite is summarised in Table 2 for all of the observational data
sets used for the evaluation. The individual entries of this scorecard are discussed in the subsections below.

## 4.1 Ozone (O$_3$)

The ozone concentrations simulated by CAMS e-suite and o-suite were evaluated using several surface (WMO-GAW, ESR-
L/GMD, IASOA, EEA-Airbase, CNEMC, AirNow), profile (IAGOS, Ozonesondes) and satellite total column (IASI) observa-
tions.

The comparison against surface observations over Europe (EEA-Airbase), Fig. 1, shows that the MNMB of the e-suite is
higher than the o-suite by up to 15% especially over Central and Northern Europe, changing some smaal underestimates in
small overestimates. The e-suite temporal correlation improved slightly compared to the o-suite, but not everywhere. The bias
in the e-suite control run is slightly lower and improved compared to the o-suite control run, in agreement with comparisons
with WMO-GAW and ESRL/GMD observations.

The surface ozone validation using observations from the China National Environmental Monitoring Center, Fig. 2, shows
that both the e-suite and e-suite control runs reduce the negative bias observed for the o-suite and o-suite control runs over the
north-eastern region extending from Shanghai to Beijing. A slight improvement is also found over the megacity of Guangzhou
in the south. This indicates less titration of ozone due to improved NO$_2$ in the e-suite/e-suite control run, see also section 4.3
below, which is likely linked to the anthropogenic emission update. The performance of the e-suite remained almost similar to
that of o-suite in central and western China. The correlation on average is about 0.75, similar for all experiments.

The evaluation of surface ozone using the AirNow ground-based stations over North America, Fig. 3, shows that the over-
estimation is slightly higher in the e-suite than in the o-suite (MNMB from +13% to +22%) while the temporal correlation (R)
and FGE improved, especially for the control run. The higher ozone bias may be related to a slight underestimation of NO$_2$ in
the e-suite compared to the o-suite (Eskes et al., 2023b). The differences between the assimilation and control runs demonstrate
the impact of the assimilation.

Surface ozone from the e-suite and o-suite was also compared to WMO-GAW, ESRL/GMD and Arctic IASOA surface
station observations (Eskes et al., 2023b). During October 2022 to February 2023 there was no significant change in the surface
ozone bias between CAMS e-suite and o-suite, with relative biases of less than -40% for most of the stations. Exception is the
SPO station over the Antarctica, where the e-suite shows a higher negative bias (-50%) than in o-suite (-40%).





The e-suite and o-suite have been evaluated with IAGOS aircraft measurements of tropospheric ozone. IAGOS L1 data is used for ozone and CO (next section), which is filtered NRT data manually validated by the PI. The time series of the daily profiles at the Frankfurt airport show that the e-suite and the o-suite have a similar performance in general (Fig. 4). However, in the free troposphere (between 350 hPa and 850 hPa) the e-suite control run shows a reduced bias compared to the other runs from both the e-suite and the o-suite from the beginning of the evaluation period until April (Eskes et al., 2023b). For

May-June, the e-suite control run develops a negative bias and the results of the other e-suite and o-suite runs are similar. This is more clearly depicted in the time series of the monthly scores by vertical layers at Frankfurt, which shows a better performance (MNMB and FGE) in the e-suite control run in the free troposphere in January-April and the degradation for May and June. The two controls runs show a seasonal difference, which is compensated for by the assimilation. At most airports worldwide the bias in the lower troposphere (pressure > 850 hPa) is slightly larger for the e-suite than for the o-suite, and in particular

over airports located in Western Africa and Eastern Asia (not shown). Conversely in the free troposphere, the bias is smaller in the e-suite that in the o-suite for most visited airports (Eskes et al., 2023b). Moreover, as regards Western Africa the bias is for most airports negative for the e-suite runs while it is positive for the o-suite runs (Eskes et al., 2023b).

In the upper troposphere (evaluations based on flight level data with a potential vorticity below 2) ozone is overestimated and the results from the e-suite and the o-suite are very similar for all runs (Eskes et al., 2023b) except for the o-suite control

run which presents slightly larger biases than the other runs over the Northern Atlantic and North America.

Tropospheric ozone profiles have also been compared to ozone sonde observations, see Fig. 5. The northern hemisphere profile difference can be considered statistically robust (p-value for tropospheric columns is below 0.01) and little difference is seen between the o-suite and e-suite in the troposphere. The e-suite control run shows a large negative bias in the upper troposphere and lower stratosphere (below 100hPa), while almost no bias is found in the lower troposphere, in agreement with

the IAGOS comparisons. For the other regions, the number of profiles is much lower, and the differences between the o-suite and e-suite are mixed: in the Tropics and Antarctica, the bias in the lower troposphere is smaller for the e-suite, while the opposite is found in the Arctic and southern hemisphere mid-latitudes. The spread in the differences is similar for both runs. Note that the ozone sonde results in the troposphere, including the change in bias around March-April, are consistent with the IAGOS plots shown above. Note that there is no significant change in the number of sondes around this time.

The global maps of monthly mean total column ozone for October 2022 is compared with satellite observations from IASI, see Fig. 6 top row. Overall, the performance is somewhat improved in the e-suite, with slightly better distributions of the ozone columns compared to the o-suite. A bias feature over the tropical Pacific ocean in October 2022 has disappeared in the e-suite. This feature may be attributed to the issue with the background covariances in Cy47R3, which was solved on 15 December. The positive bias at low latitudes in October is smaller in the e-suite than in the o-suite. For December 2022, the difference

between the e-suite and the o-suite is less prominent. The positive bias above the Pacific ocean is improved in the e-suite and remains mostly below 5%. There is hardly a difference between e-suite and o-suite for June 2023.





**Figure 1.** Spatial distribution of the O₃ comparisons for Europe, for the e-suite (first row, left: MNMB, right: R), the o-suite (second row, left: MNMB, right: R), the e-suite minus the o-suite differences (third row, left: FGE, right: R) and the e-control minus the o-control differences (fourth row, left: FGE, right: R) for the period 2023-10-01 to 2023-06-27.






**Figure 2.** Spatial distribution of the O₃ comparisons for China, for the e-suite (first row, left: MNMB, right: R), the o-suite (second row, left: MNMB, right: R), the e-suite minus the o-suite differences (third row, left: FGE, right: R) for the period 2023-10-01 to 2023-06-27. The e-control minus the o-control display similar differences as e-suite minus o-suite (not shown).





**Figure 3.** Spatial distribution of the $O_3$ comparisons for the USA and Canada, for the e-suite (first row, left: MNMB, right: R), the o-suite (second row, left: MNMB, right: R), the e-suite minus the o-suite differences (third row, left: FGE, right: R) and the e-control minus the o-control differences (fourth row, left: FGE, right: R) for the period 2023-10-01 to 2023-06-27.



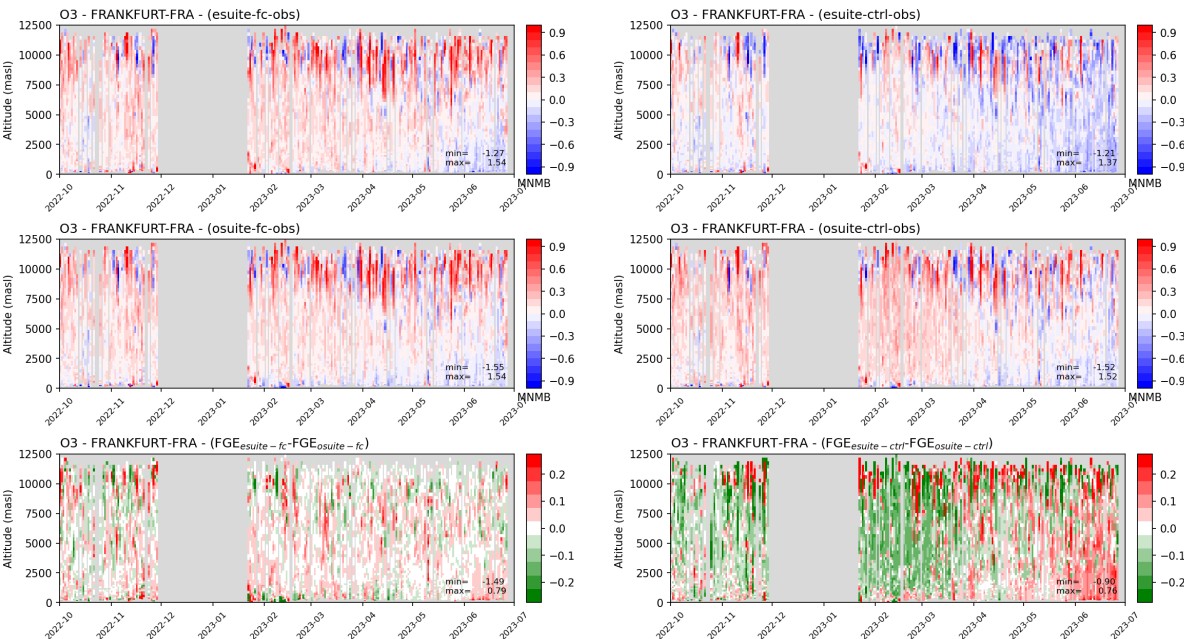

**Figure 4.** Ozone comparisons using aircraft profile observations from IAGOS (http://www.iagos.org). Time series of modified normalized differences CAMS-IAGOS in the daily profiles of ozone at Frankfurt between October 2022 and 27 June 2023 for the e-suite (top-left), the o-suite (middle-left) and their respective control runs (right). The fractional gross error differences between the e-suite and the o-suite are shown in the bottom row.



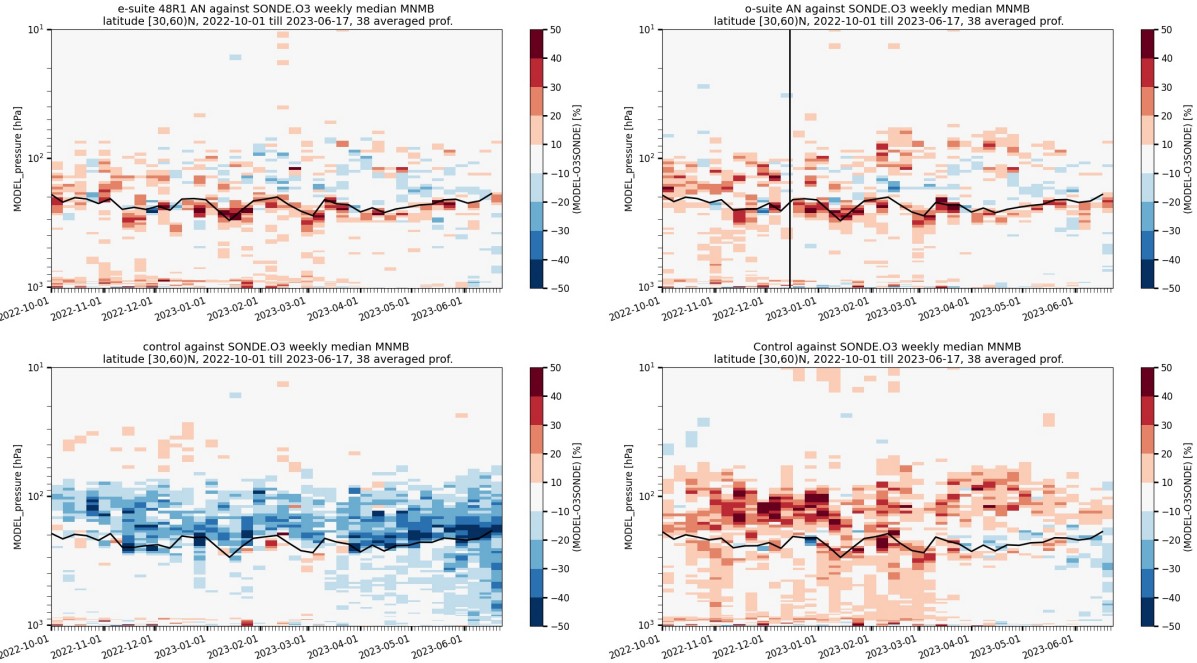

**Figure 5.** Comparisons against ozone sondes. Time-pressure curtain plot of ozone profile MNMB for the e-suite analysis (top-left), the o-suite analysis (top-right), the e-suite control run (bottom-left) and o-suite control run (bottom-right) against sonde profiles for the northern hemisphere midlatitudes. The horizontal black line represents the tropopause. The vertical black line in the o-suite plot indicates the 15 December change in the background error covariancein Cy47R3.





**Figure 6.** Global maps of monthly mean total column ozone (Dobson Unit) compared with satellite observations from IASI. The plot shows the e-suite result (top left) and the IASI observations (top right) for October 2022, below the relative bias of the e-suite (left column) and the relative bias of the o-suite (right column) with respect to IASI (%) for October 2022 (second row), December 2022 (third row) and June 2023 (bottom).



## 4.2 Carbon Monoxide (CO)

The simulated carbon monoxide of CAMS e-suite and o-suite was evaluated using surface observations from the WMO-GAW, EEA-Airbase, CNEMC and AirNow networks, vertical profiles from IAGOS aircrafts and NDACC FTIR measurements, and satellite total column retrievals from MOPITT and IASI.

Over China, the comparison with surface CO observations indicates an overall reduction in CO concentrations in the e-suite, which leads to a reduced positive bias and better performance in the megacities over north-eastern China and particularly over Shanghai, Hangzhou and Beijing (Fig. 7). The negative bias and the FGE increased in the e-suite in the rest of China, including central China. The e-suite shows similar correlations to the o-suite with values exceeding 0.6 over most stations. The reduction in CO can be linked to the emission update in Cy48R1.

Compared to surface CO observations from five WMO-GAW stations (Hohenpeissenberg, Jungfraujoch, Sonnblick, Zugspitze, and Monte Cimone, located in Europe) there was no significant change in the bias between CAMS e-suite and o-suite runs, which does not exceed ±10% in general (Eskes et al., 2023b). The correlation for the e-suite has slightly improved compared to the o-suite over Europe and the Cape Verde station in the Tropics.

The comparison of the CO mixing ratios against EEA-Airbase observations over Europe shows that e-suite perform better than o-suite in terms of bias and correlation over most stations (Eskes et al., 2023b). In all cases note that e-control scored better than o-control as well, hence the detected improvements at the surface mainly originate from the model changes rather than assimilation changes.

The e-suite has been evaluated with IAGOS measurements of CO (Level 1 data) at different airports. The time series of the daily profiles (curtain plots) of the MNMB at the Frankfurt airport are presented in Fig. 8. The e-suite shows a smaller bias than the o-suite at all altitudes for the analysis and the 1-day forecast. This is also shown in the time series of the monthly scores by layers at Frankfurt, with an analysis and forecast improvement for the e-suite in both the Lower Troposphere (LT) and the Free Troposphere (FT) (Eskes et al., 2023b). However, CO is still underestimated by both the e-suite and the o-suite with a larger bias in the LT than in FT. According to the fractional gross error (FGE) monthly values, the e-suite improvement over the o-suite is about -0.05 for most of the months in the LT against about -0.02 in the FT, while for correlation results, no notable difference is found (Eskes et al., 2023b). The control runs from the e-suite and o-suite show a different seasonal pattern of the bias with a notable increase of the bias starting in early spring for both models. Like for the the assimilated runs, the bias from the control run is smaller for the e-suite than from the o-suite but only until March. This can also be seen in the monthly scores time series (Eskes et al., 2023b). For the remaining months the performance is similar for both control runs in the lower troposphere and slightly better for the o-suite control run in the free troposphere. For all other airports worldwide the bias (MNMB) in the low troposphere is very similar for the e-suite and the o-suite. However and like at Frankfurt, at most visited airports the absolute differences (FGE) are better in the free troposphere for assimilated runs of the e-suite compared with those of the o-suite (Eskes et al., 2023b). Regarding the control run results, the results are similar for both models. In the upper troposphere, at cruise altitude, CO is underestimated by both the e-suite and the o-suite runs. Like in the free troposphere





**Figure 7.** Spatial distribution of the CO surface comparisons for China, for the e-suite (first row, left: MNMB, right: R), the o-suite (second row, left: MNMB, right: R), the e-suite minus the o-suite differences (third row, left: FGE, right: R) for the period 2023-10-01 to 2023-06-27. The e-control minus the o-control display similar differences as e-suite minus o-suite (not shown).



there is a clear improvement of the bias in the upper troposphere over all regions in the e-suite compared to the o-suite and in particular for the analysis (Eskes et al., 2023b).

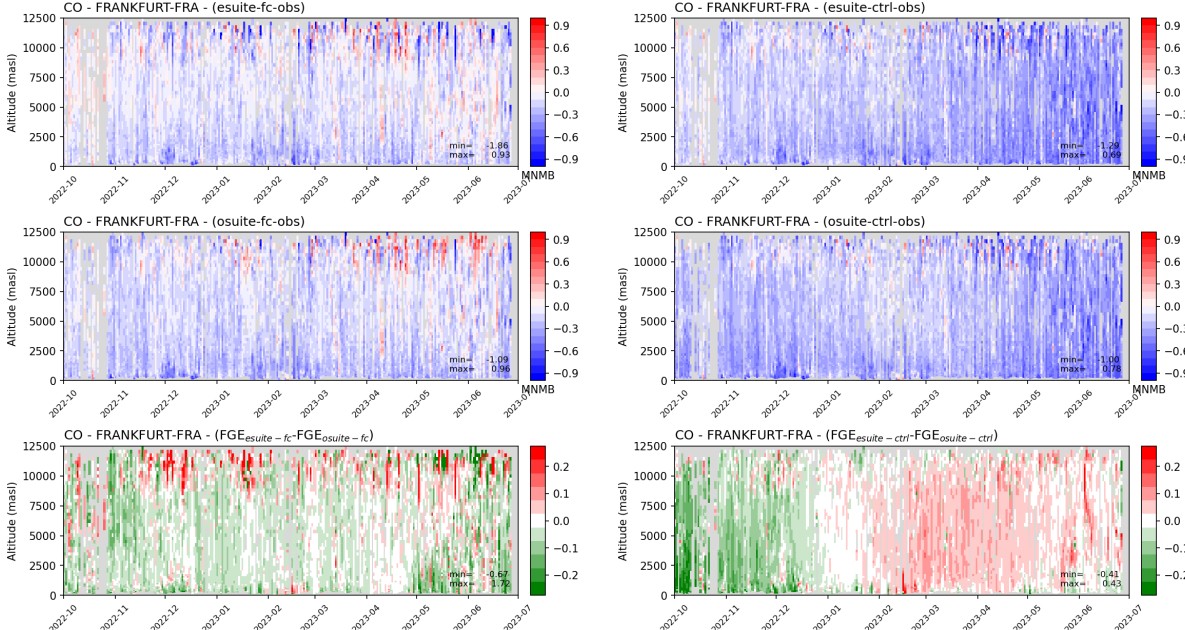

**Figure 8.** CO comparisons using aircraft profile observations from IAGOS (http://www.iagos.org). Time series of modified normalized differences CAMS-IAGOS in the daily profiles of ozone at Frankfurt between October 2022 and 27 June 2023 for the e-suite (top-left), the o-suite (middle-left) and their respective control runs (right). The fractional gross error differences between the e-suite and the o-suite are shown in the bottom row.

The results of the comparison with satellite CO column retrievals from the IASI and MOPITT instruments are shown in Fig. 9. Averaging kernels have been used in these comparisons. The IASI total column CO observations are well reproduced by the CAMS e-suite in terms of absolute amounts and spatial distribution. The e-suite relative bias with reference to IASI stays

mostly within 20%, with wide areas below 5% and some negative bias above the Southern Africa biomass burning area (up to 20%) for December 2022, and a positive bias south of 60°S. The e-suite performs much better than the e-control, which shows negative biases for the major part of the globe up to 20% regularly and peak amounts above South Africa exceeding 40%. The o-suite has mostly a negative bias compared to IASI, up to 20% regularly and up to 30% over the Pacific. Rows four and five show the e-suite relative biases for March 2023 and June 2023, that support the previous findings also for the other seasons. The

MOPITT comparison looks very different, as overall biases are smaller for the o-suite than for the e-suite. The e-suite shows widespread positive biases up to 20% which is not present in the o-suite. Only the negative biases of the o-suite up to 20% above Northern Africa improved to remain mostly below 10% in the e-suite. Similar observations are made for the additional months from different seasons. These results for CO total columns demonstrate changes in the bias correction implemented in





the e-suite, which is now using IASI-C and TROPOMI as reference. Note, that TROPOMI CO data were not assimilated in the
e-suite until 28th of April, so the impact of TROPOMI has not been explicitly evaluated here.

Comparisons were also made against CO partial columns and profiles from the FTIR instruments, part of the surface remote
sensing NDACC network (Fig. 10). For tropospheric CO columns, the bias for the e-suite is reduced compared to the o-suite
for almost all sites. Although correlations are similar for the o-suite and e-suite, the ratio of the standard deviation in the
troposheric columns for the e-suite and the FTIR time series is higher compared to the o-suite. The bias at the tropical sites
has switched sign and is now positive. For stratospheric CO columns the bias for the e-suite is reduced significantly for the
southern hemispheric sites and is now of the order of the measurement uncertainty. The southern hemisphere correlations are
also higher for the e-suite (in particular for the Antarctic site Arrival Heights). For the northern hemisphere, the e-suite and o-
suite perform similarly. The o-suite has a degraded performance during this period in the southern hemisphere (Wollongong and
Lauder stations, where the tropospheric columns show an increased negative bias compared to the months before October 2022
and the stratospheric columns show a strong ($+20\%$) positive bias. A direct comparison between the o-suite and e-suite should
therefore be interpreted with care. The figure shows that for the e-suite a reduced positive bias remains in the stratosphere for
this period. This can be considered an improvement compared to the o-suite performance prior to October 2022 (negative bias
$< -20\%$) and past October 2022 (positive bias $> +20\%$).

As of 22 May 2023, wildfires have been raging across Canada's Alberta province for three weeks. IAGOS aircraft profiles
of CO over Montreal and Calgary picked up the pollution plumes from these events, and have been compared with the CAMS
results (Eskes et al., 2023b). For most profiles studied during this event, the e-suite is performing better than the o-suite with
values of the CO mixing ratios higher than those provided by the o-suite, although the forecast model remains to face difficulties
to capture the high concentrations in the plumes, while the altitude is simulated with success.



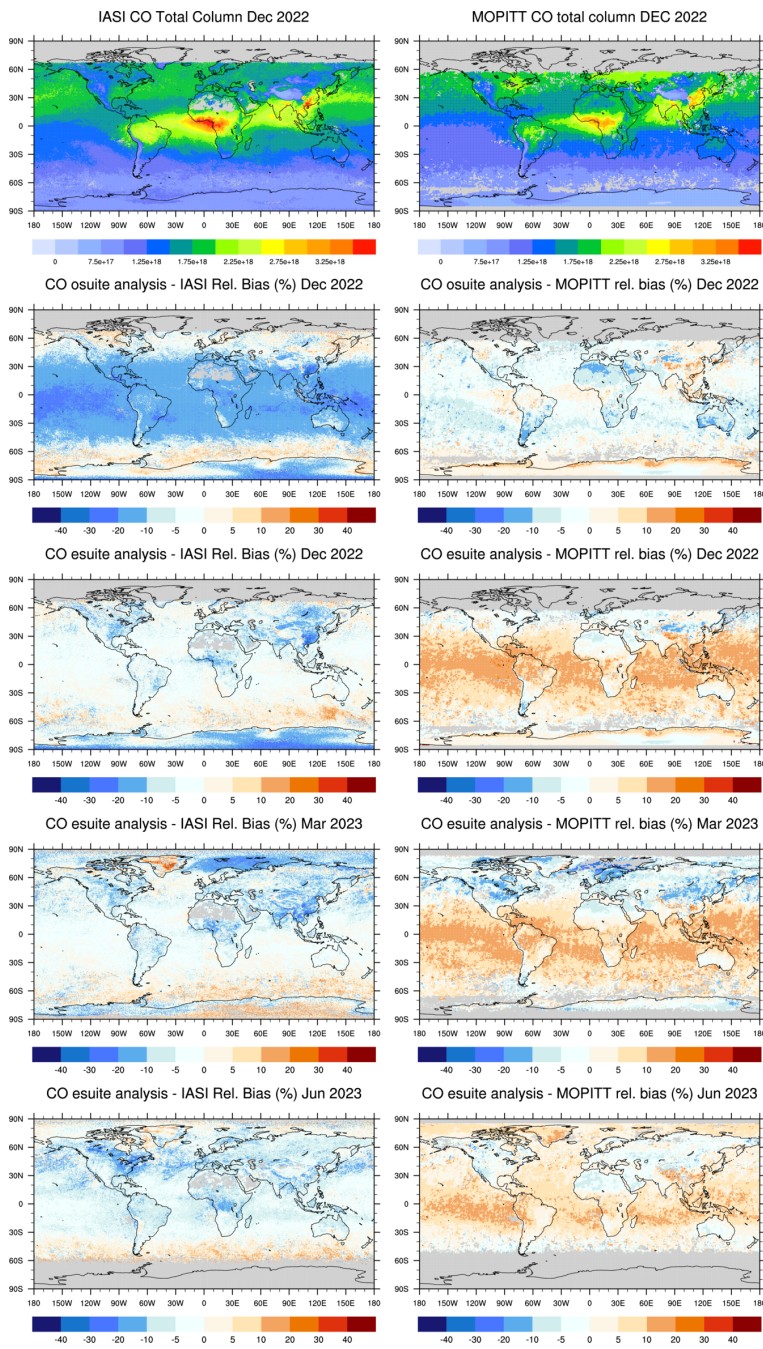

**Figure 9.** Global maps of monthly mean total column CO comparing satellite observations from IASI (left column) with MOPITT (right column) validation including the satellite observation (first row), the relative bias of the o-suite result with respect to the satellite data (second row), and the relative bias of the e-suite result (third row) for Dec 2022, as well as the e-suite relative bias for March and June 2023 (fourth and fifth row, respectively).





**Figure 10.** Curtain plots of the CO profile relative difference of the CAMS products compared to FTIR observations from the NDACC network for the two southern hemispheric stations Lauder (left) and Wollongong (right). Shown are the differences with the e-suite (top row), o-suite (second row), e-control (third row) and o-control (bottom row). Model profiles are smoothed with the FTIR averaging kernels. The horizontal black line is the tropopause. The vertical black line indicates 15 December, when the change in the background error covariance was implemented.



### 4.3 Nitrogen dioxide (NO$_2$)

The CAMS configurations were also compared with surface observations in China, Europe and North America.

Over China (Fig. 11), the e-suite and e-suite control run significantly reduce the Cy47R3 positive offset in surface NO$_2$ over most regions in eastern China , which also led to a reduced bias in surface ozone as discussed in section 4.1. The correlation of the e-suite/e-suite control run is similar (slightly reduced) compared to the o-suite/o-suite control run. The mean correlation coefficient averaged over all stations is 0.52 for the e-suite and 0.54 for the o-suite.

The comparison with surface NO$_2$ observations in Europe obtained from EEA-Airbase shows that the e-suite performs better than o-suite in terms of bias for a majority of the stations especially in Central and Northern Europe (Eskes et al., 2023b). Interestingly, this coincides with the regions where a slightly higher bias was found for ozone in the e-suite indicating a chemical regime changes in the model. In terms of correlations the e-suite and o-suite performed almost equally.

The evaluation of surface NO$_2$ using the AirNow ground-based stations over North America shows that the underestimation
in e-suite/e-suite control run is higher than o-suite/o-suite control run (from about -30% to -40%) while the temporal correlation (R) slightly improves (from 0.49 to 0.54) (Eskes et al., 2023b).

The CAMS e-suite Cy48R1 tropospheric NO$_2$ column data is compared to the TROPOMI scientific IUP Bremen tropospheric NO$_2$ product, see Figure 12. Three different months are selected for performance demonstration, including those periods with highest and lowest solar elevation for the two hemispheres. Overall, the e-suite results correspond well with
365 the observational data in terms of spatial and temporal variations and absolute amounts. However, there are also apparent differences between model and measurements. Largest differences are mainly observed above regions with major emissions. Locations strongly affected by anthropogenic pollution such as hotspots in China can show strong positive biases partly above 100%. In comparison to the o-suite, however, the e-suite performs better compared to TROPOMI especially above strongly polluted regions such as Eastern China. Here, the positive bias in the o-suite is reduced in the e-suite as well as above other
hotspot regions. This effect is best visible in the December maps (left column of Fig. 12). This improvement is likely due to the updated emissions used in Cy48R1. Most areas with strong positive biases are smaller in spatial extent in the e-suite compared to the o-suite, and smaller in their bias values. In addition, there are a few confined regions, where the e-suite shows positive biases not present in the o-suite comparison such as above the North Sea. This can mainly be explained by the lower anthropogenic emissions in CAMS-GLOB-ANT v5.3 as applied in the e-suite, compared to CAMS-GLOB-ANT v4.2 in the
o-suite.

Some typical biomass burning areas such as Southern Africa or South America show negative biases around 40% compared to TROPOMI. Negative biases are seen also above parts of the US, the Northern Atlantic, Northern Pacific and parts of Asia. In contrast, positive biases are found over boreal fires. No noteworthy degradation in performance is seen when moving from the previous o-suite to the e-suite results. This is also true concerning the background regions, both e-suite and o-suite perform
well in representing background values close to zero. The negative biases for some oceanic background regions are slightly larger for the e-suite than for the o-suite. The negative biases above the oceans on the Northern Hemisphere show up in areas




**Figure 11.** Spatial distribution of the NO$_2$ surface comparisons for China, for the e-suite (first row, left: MNMB, right: R), the o-suite (second row, left: MNMB, right: R), the e-suite minus the o-suite differences (third row, left: FGE, right: R) for the period 2023-10-01 to 2023-06-27. The e-control minus the o-control display similar differences as e-suite minus o-suite (not shown).



where the absolute tropospheric $NO_2$ column is small and close to background levels. Slight differences in absolute amounts hence cause comparably large relative bias values. Overall, the performance of the e-suite is slightly better than the o-suite.



**Figure 12.** Monthly mean global maps of tropospheric $NO_2$ column densities for TROPOMI satellite observations (first row), the e-suite (second row), the difference between e-suite and satellite (third row), the relative difference between e-suite and satellite (fourth row) and the relative difference between o-suite and satellite (fifth row) for December 2022 (left column), March 2023 (middle column) and June 2023 (right column). Units: 1e15 molecules $cm^{-2}$. Note that for the relative bias the regions with background values below $5 \times 10^{14}$ molec. $cm^{-2}$ are not included in the analysis.



## 4.4 Sulfur dioxide ($SO_2$)

The comparison against $SO_2$ surface observations from the China National Environmental Monitoring Center shows that e-suite/e-suite control run significantly reduces the positive bias observed for the o-suite/o-suite control run over most regions in eastern China as indicated by the difference in FGE between e-suite and o-suite (Fig. 13. However, a high overestimation (>50%) is still observed in the e-suite for many of the stations, indicating a remaining possible overestimation of $SO_2$ emissions in the updated CAMS_GLOB_ANT inventory. Note that the temporal correlation (R) is slightly reduced in the e-suite compare

to the o-suite.

The comparison of the $SO_2$ mixing ratio against EEA-Airbase observations over Europe shows that the e-suite/e-suite control run performs better than the o-suite/o-suite control run in terms of bias for most stations (Eskes et al., 2023b). The correlations are almost equal.





**Figure 13.** Spatial distribution of the $SO_2$ surface comparisons for China, for the e-suite (first row, left: MNMB, right: R), the o-suite (second row, left: MNMB, right: R), the e-suite minus the o-suite differences (third row, left: FGE, right: R) for the period 2023-10-01 to 2023-06-27. The e-control minus the o-control display similar differences as e-suite minus o-suite (not shown).



## 4.5 Formaldehyde (HCHO)

The tropospheric formaldehyde (HCHO) column from the CAMS e-suite is compared to the TROPOMI scientific IUP Bremen tropospheric HCHO product (Eskes et al., 2023a). Monthly mean global maps and differences are shown in Fig. 14.

The e-suite overestimates the tropospheric HCHO columns over South America and Northern Australia by more than 60% in December and March, and there is also an overestimation over Indonesia. The allocation of the HCHO columns over Northern and Southern Africa in the two months is well reproduced by the model, but there are biases in certain time periods, such as

the overestimation North of the equator and an underestimation in the South for June 2023.

In comparison to the o-suite, the overestimation above South America and Northern Australia is stronger in the e-suite visible in the relative biases of the e-suite and the o-suite in row four and five of Fig. 14. This could be the effect of the updated isoprene chemistry, which induced a higher HCHO production. Above Northern Africa, the HCHO values show a larger positive bias, while over Southern Africa the negative bias is slightly less pronounced in the e-suite. Overall, the performance is similar, but

slightly degraded for the e-suite.







**Figure 14.** Monthly mean global maps of tropospheric HCHO column densities for TROPOMI satellite observations (first row), the e-suite (second row), the difference between e-suite and satellite (third row), the relative difference between e-suite and satellite (fourth row) and the relative difference between o-suite and satellite (fifth row) for December 2022 (left column), March 2023 (middle column) and June 2023 (right column). Units: 1e15 molecules cm$^{-2}$. Regions with mean background values below $3 \times 10^{15}$ molec/cm2 are excluded from the analysis (white areas).





## 4.6 Stratospheric ozone (O$_3$)

Stratospheric ozone from CAMS e-suite Cy48R1 and o-suite have been compared against satellite limb profiles and ozoneson-des, considering the analyses, the 5th day forecasts and the o-suite control run, see Fig. 15 and (Eskes et al., 2023b). The full e-suite period is considered, from 1 October 2022 to 27 June 2023).

The main conclusions are:

- In the lower stratosphere, at pressures higher than 10 hPa, the e-suite agrees better with the observations than the o-suite, with lower biases, lower FGE, and higher correlations for the 9-month period evaluated.

- The e-suite agrees much better with observations than the o-suite in the tropics.

- The e-suite 5-day forecast agrees better with independent observations than the o-suite during ozone hole conditions.

- In the upper stratosphere (at pressure lower than 10 hPa), the e-suite displays a larger negative bias than the o-suite, while the correlations are improved in the e-suite in the upper stratosphere in the tropics.

- The control run of the e-suite shows a higher negative bias than the o-suite control run in the upper stratosphere above the 10 hPa level.

The difference between the o-suite and the e-suite are largely due to the model update, which is demonstrated by comparing
the control runs of the o-suite and the e-suite.

Stratospheric ozone has also been compared to lidar profile observations from the NDACC network for the full e-suite period of 9 months, see Fig. 16. Above 10hPa the e-suite mean profile deviates from the o-suite, showing a negative bias compared to the lidar observations, increasing with altitude. Between 50hPa and 10hPa the e-suite overestimates the ozone concentration. The negative bias feature observed around 20 hPa in the o-suite at Mauna Loa is no longer present in the e-suite.
The dispersion (std) in the differences is slightly, but systematically, improved for the e-suite. In general, the conclusions for the lidar comparisons are very similar to the satellite comparisons.



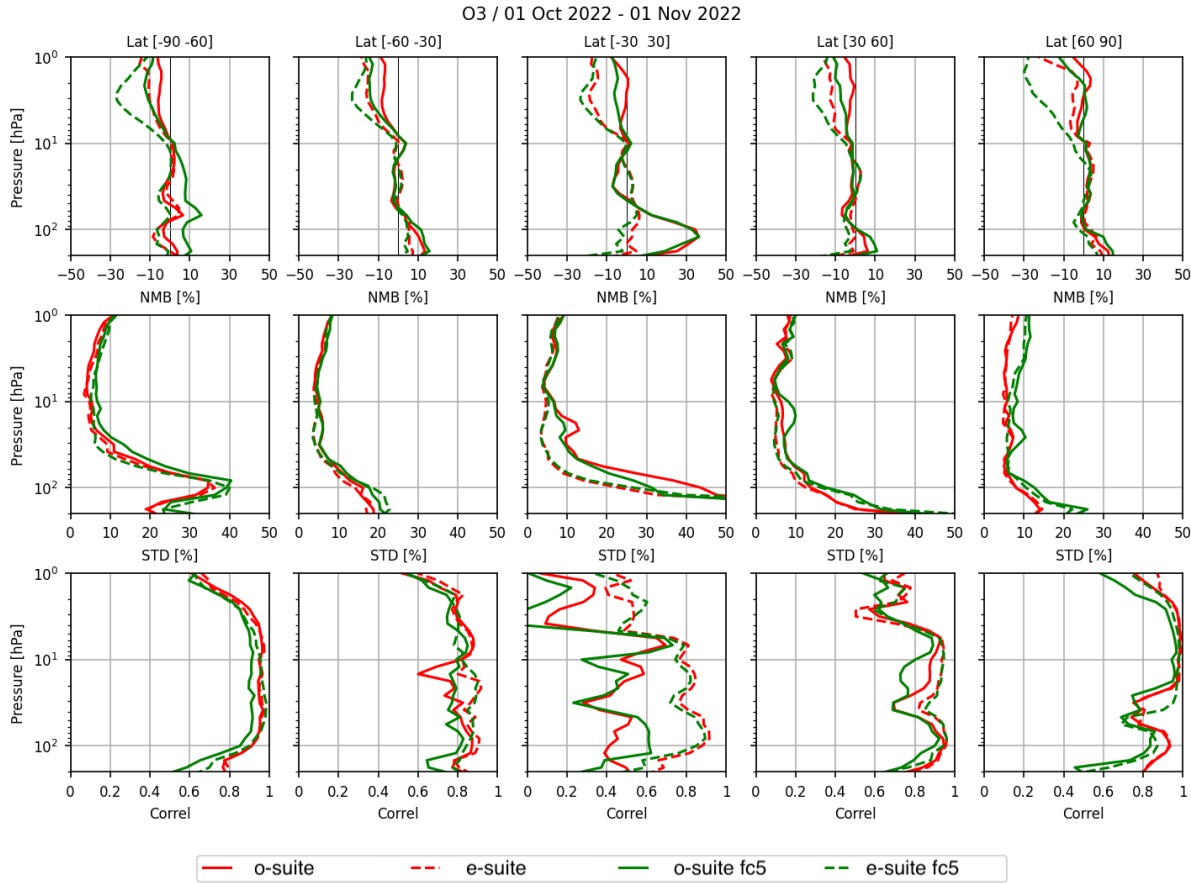

**Figure 15.** The e-suite and o-suite compared to a Multi-Instrument-Mean (MIM), consisting of observed ozone profiles from ACE-FTS v4.1, Aura-MLS v4.2, OMPS-LP v2.5, SAGE-III/ISS v5.2 and ozonesondes. Top row: normalised mean difference between the MIM ozone observed profiles and the o-suite analyses (solid red line), the o-suite 5th day forecast (solid green line), the e-suite analyses (dashed red line) and the e-suite 5th day forecast (dashed green line). The figure refers to the period October 2022, which is during the Antarctic ozone hole conditions. Five latitude bands are considered, from left to right: 90°S-60°S, 60°S-30°S, 30°S-30°N, 30°N-60°N and 60°N-90°N. The corresponding standard deviation of the differences and the correlation coefficient between observed profiles and the e-suite and o-suite runs is shown in the second and the third row respectively.





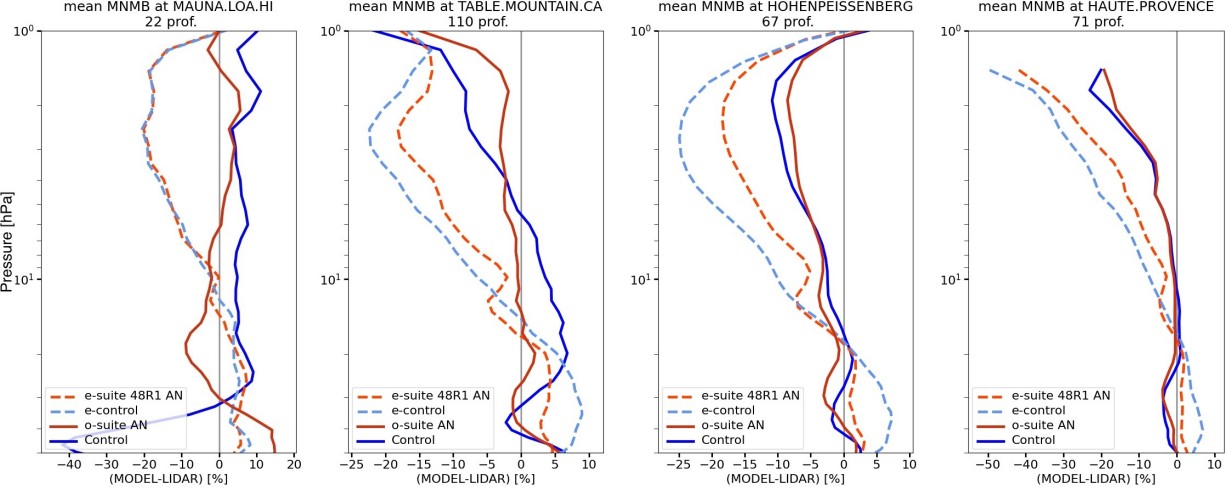

**Figure 16.** Mean ozone profile differences for e-suite (red dash), o-suite (red solid), e-suite control run (blue dash) and o-suite control run (blue solid) compared to stratospheric ozone lidar observations of four stations from the NDACC network.



## 4.7  Other trace gases in the stratosphere

Other species simulated in the stratosphere in the e-suite have been compared with satellite limb profiles considering 3 hourly first day forecast. The new CAMS cycle 48R1 introduces full stratospheric chemistry, and meaningful comparisons with the

o-suite is not possible. But reasonable concentration of these species are important because of their impact on ozone. Species considered are: $CCl_4$, $CFC-11$, $CFC-12$, $HCFC-22$, $ClO$, $ClONO_2$, $HCl$, $CH_4$, $H_2O$, $HNO_3$, $N_2O$, $N_2O_5$, $NO_2$, NOx $(NO+NO_2)$, and $O_3$.

There is a relatively good agreement between the e-suite and observations for long-lived species ($CCl_4$, $CFC-11$, $CFC-12$, $CH_4$, $HCFC-22$ and $N_2O$) and $HCl$ between 10 hPa and 200 hPa, see Figure 17 (Eskes et al., 2023b). The chemical $H_2O$

tracer in the CAMS e-suite shows a negative bias. Even though not perfect, CAMS e-suite $NO_2$ (and NOx) agree relatively well with observations, which was not the case in the previous o-suite cycles. Above the 10 hPa level the concentrations are overestimated compared to ACE-FTS, and there is room for improvement. For $ClO$, the agreement is good against MLS, less good with ACE-FTS, but this dataset is supposed to be less reliable than MLS for this species.

Stratospheric $NO_2$ columns have been compared with TROPOMI retrievals, see Fig. 18. The observational $NO_2$ strato-

spheric column is represented well by the e-suite in terms of absolute amounts, latitudinal variation and temporal changes. Over the largest part of the globe, column amounts agree to within $2 \times 10^{15}$ molec/cm$^2$, deviating by less than 10%. The e-suite performs well in reproducing the general strong increase in stratospheric $NO_2$ at high latitudes in the summer hemisphere.





**Figure 17.** Comparison of $O_3$, $H_2O$, $NO_2$, NOx, $CCl_4$, CFC$-$11, CFC$-$12 from the e-suite (first day forecast) with ACE-FTS observations between 1 Oct 2022 - 27 June 2023. Five latitude bands are considered, from left to right: 90°S-60°S, 60°S-30°S, 30°S-30°N, 30°N-60°N and 60°N-90°N.



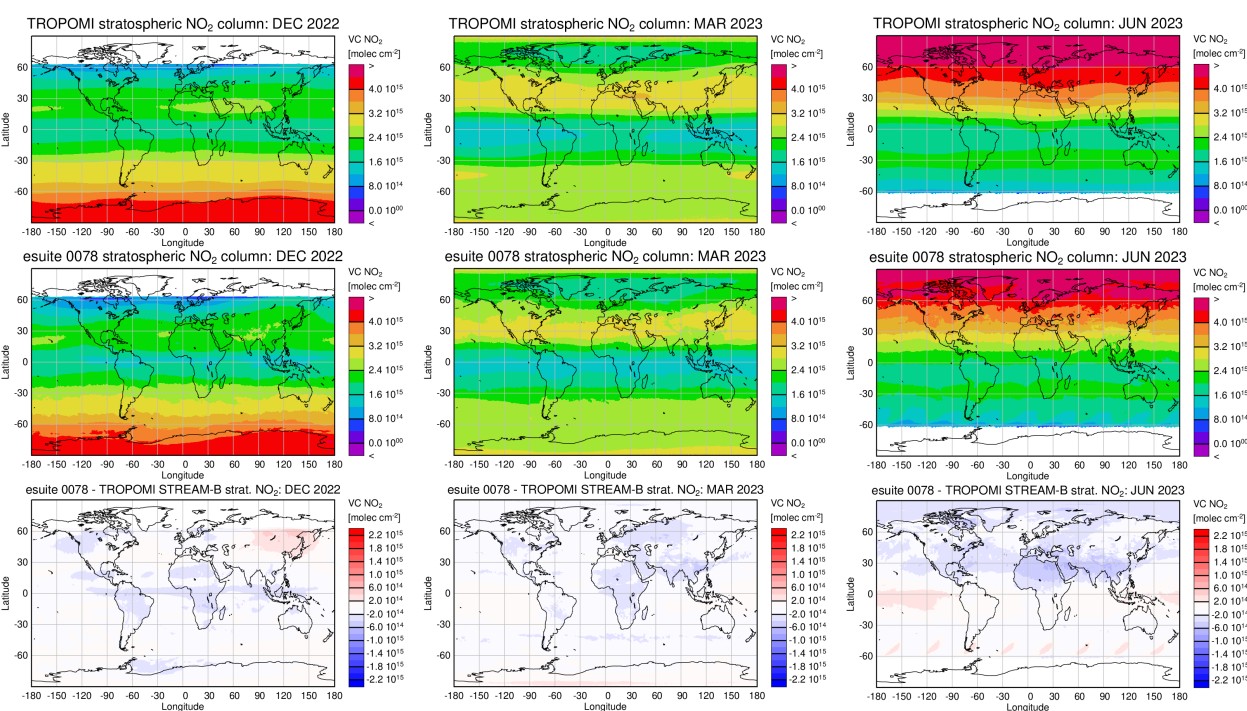

**Figure 18.** Monthly mean NO$_2$ stratospheric column amounts retrieved using TROPOMI observations (top) compared to the e-suite results (second row). The third row shows the differences between the e-suite and TROPOMI. Results are shown for December 2022 (left column), March 2023 (second column), and June 2023 (right column). The STREAM-B algorithm was used to estimate the stratospheric contribution to the total retrieved column (Eskes et al., 2023a).





## 4.8 UV radiation

While no changes to the UV code were implemented in Cy48R1, the surface UV index (UVI) forecasts are affected by the

changes in the optical depth of trace gases (of which most notably stratospheric ozone), aerosols, and thick clouds. The net effect of these factors on the hourly forecast performance of the new model cycle were estimated using ground based UVI measurements as the reference data at the 38 stations presented in (Eskes et al., 2023b), located in Europe, Israel, Thailand, Australia, New Zealand, and the Antarctic.

Figure 19 illustrates the statistical improvement of Cy48R1 compared with Cy47R3 when evaluated against the ground

based observations in terms of MNMB, FGE, and R. MNMB and FGE included all available hourly UVI forecasts, while R included only UV forecasts close to local noon in order to emphasise the importance of atmospheric composition (instead of solar zenith angle SZA), and to highlight the time of day when UV radiation is typically most intense and hazardous. Overall, the statistics (mean MNMB=5 %, mean FGE=0.30, and mean R=0.82) indicate that CAMS UV forecasts are of good quality in Cy48R1. The mean values of R and FGE indicate no significant changes in correlation and scatter between the old and the

new model cycles, however the mean MNMB increases slightly from 2 % to 6 %. This finding is supported by comparing the zonal mean of daily maximum UVI between the two model cycles, which increased by +0 to +4 % with the largest increase +2 to +4 % occurring between 50°S and 60°N in Cy48R1 compared to Cy47R3.





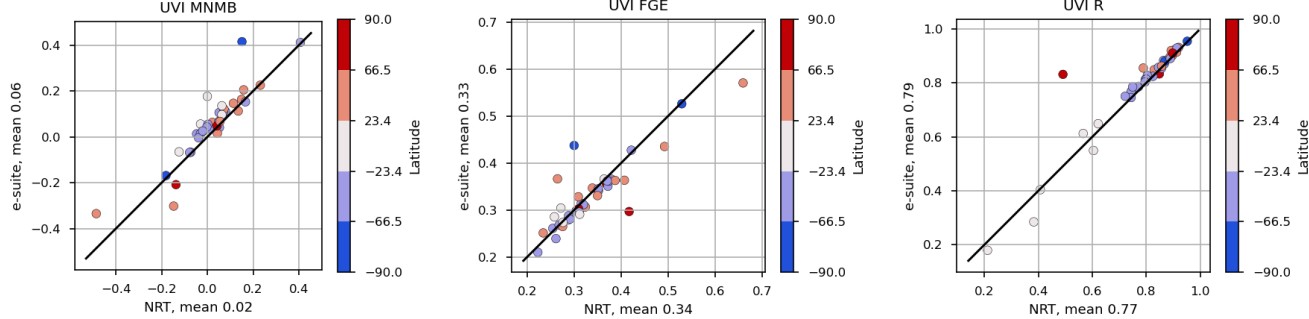

**Figure 19.** Hourly UV index values from the CAMS e-suite and o-suite evaluated against ground based UV measurements from 38 stations located in Europe, Israel, Thailand, Australia, New Zealand, and the Antarctic. Scatterplots of MNMB, FGE and R between measurements, and o-suite (Cy47R3, x-axis) and measurements, and e-suite (Cy48R1, y-axis). Each dot represents a single measurement station, either in the northern latitudes (red) or southern latitudes (blue). The time range is from 2022-10-01 to 2023-06-27.

## 4.9 Aerosol

The Cy48R1 introduces a redistribution of aerosol optical depth per species (Fig. 20 first column). Starting from Cy48R1, the secondary organic aerosol (SOA) optical depth is now provided separately, whereas prior to this update, the SOA optical depth was included as part of the organic matter (OM) optical depth. For this report, the OM optical depth in the e-suite includes the SOA, in order to ensure comparability with the OM optical depth in the o-suite. Compared to o-suite there is less AOD for the e-suite particularly due to reduction in sulphate optical depth (Fig. 20 second column). For the e-suite there is less black carbon optical depth, particularly over Central Africa. Nitrates optical depth slightly decrease over southeast Asia and Sahel. Contrarily, the sea salt and ammonium optical depth increases while dust and organic matter changes are regionally dependent. The Ångström Exponent (AE) is considerably lower globally (reduction by -0.12) in e-suite, especially over Middle East and Sahara, while it increased over South Africa, South America and Australia.

These changes are related to (1) the introduction of two new secondary organic aerosol tracers (anthropogenic and biogenic) along with their respective precursor gas tracers (2) modifications to dust emissions and removal simulation that increased the global dust mass burden by a factor of 2, (3) a review of aerosol optical properties for dust and brown carbon, as well as (4) improvements on secondary inorganic aerosol simulation. Note that the AOD and AE mostly increases due to assimilation, except dust (Fig. 20 third column).

The evaluation of daily Aerosol Optical Depth (AOD) and Ångström Exponent (AE) against the ground based network of AERONET version 3 level 1.5 stations shows that e-suite overestimates less compared to o-suite for both parameters (Fig. 21 and Fig. 22). The e-suite improves in terms AOD MNMB globally (from +20.7% in o-suite to +7.3% in e-suite), especially over North America, Europe, East Asia and Middle East. Over South-East Asia and Sahara the MNMB performance remains almost unchanged. Notably for South Asia a deterioration in performance (higher underestimation) is observed compared to





the o-suite. The correlation (R) performance remains unchanged (0.80 in both o-suite and e-suite). Note that both control runs are underestimating AOD (not shown).

The AE, a parameter which is indicative of the aerosol size distribution, improves in terms of global bias (Mean Bias from +0.28 in o-suite to +0.18 in e-suite), with regional improvements in Europe, North America as well as East China (Fig. 22). The e-suite updated dust emissions and deposition fluxes, resulting in a coarser aerosol size distribution (smaller AE) and dust mass burden over the deserts in the e-suite control run. Results specifically to dust changes are discussed in subsection 4.11. The correlation (R) of global AE remains almost unchanged (from 0.59 in o-suite to 0.62 in e-suite), unchanged in Europe and

East Asia, with a small improvement in South Asia, Sahara and Middle East and a considerable improvement over South and North America, though R still remains below 0.5 in the latter case AeroVal (2023).





**Figure 20.** AOD (1st row), AOD for each species (2nd to 8th row) and Ångström Exponent (9th row) for e-suite (1st column), differences of e-suite - o-suite (2nd column) and differences of e-suite - e-control (3rd column) for the period 2022-10-01 to 2023-06-27. Percentage at the bottom right corner of first column displays the relative contribution of each species optical depth to AOD.







**Figure 21.** Spatial distribution of the daily Aerosol Optical Depth MNMB, FGE and R for the e-suite (first row), the o-suite (second row), the e-suite minus the o-suite differences (third row) and the e-control minus the o-control differences (fourth row) for the period 2023-10-01 to 2023-06-27 using as a reference AERONET v3 level 1.5.





**Figure 22.** Spatial distribution of the daily Ångström Exponent MNMB, FGE and R for the e-suite (first row), the o-suite (second row), the e-suite minus the o-suite differences (third row) and the e-control minus the o-control differences (fourth row) for the period 2023-10-01 to 2023-06-27 using as a reference AERONET v3 level 1.5.



## 4.10    Particulate matter

Global daily near real time data of particulate matter, under 10 (PM10) and 2.5 (PM2.5) micrometres in diameter, from the surface observational networks AirNow (North America, U.S. Environmental Protection Agency (EPA)), EEA-NRT-rural (Europe, European Environmental Agency (EEA)), and CNEMC (China National Environmental Monitoring Centre) are used to evaluate e-suite and o-suite experiments for the period October 2022 to June 2023 (Figure 23). Note that all stations are considered for the AirNow and CNEMC networks, including urban stations, hence CAMS may not be able to fully capture the very high PM2.5 and PM10 measured with the observations at local urban scale due to the coarse spatial resolution of the model. We focus in this section on the evaluation results for PM2.5. The results for PM10 were found to be very similar to that of PM2.5. They are presented at the Aeroval website (AeroVal, 2023) and are discussed in Eskes et al. (2023b).

Over North America and Europe, e-suite improves PM2.5 in terms of MNMB (-3.2% and 8.9% respectively) compared to the o-suite, which exhibited a small overestimation of about +9.5% and +16.2%. Contrary over China the MNMB of e-suite (-19.9%) the e-suite exhibits a higher and negative bias when considering the performance over all available stations compared to o-suite (+3.8%). The spatial distribution of MNMB over China forms a clear dipole pattern. The eastern side, that encompass most of China's mega cities, mostly overestimates PM2.5, while in the western part, that contains less populated high altitude regions, underestimates PM2.5. The PM2.5 of e-suite outperforms o-suite in the eastern part of the country, with lower than 25% MNMB in most stations, while the opposite stand for the western part, where e-suite display more negative than -50% MNMB in most stations. The measured PM2.5 over China displays a peak in January, which is strongly underestimated by both e-suite and o-suite (Figure 20). The correlation for e-suite and o-suite remains unchanged in Europe (about 0.47), slightly improves over China (from 0.55 to 0.57) and better in North America (from 0.39 to 0.55). The reduction of sulphates (Figure 20), the reduction of the PM2.5 positive bias (Figure 23) and the reduced positive bias of $SO_2$ over China (Figure 13) are linked to the reduced $SO_2$ emissions in e-suite.





**Figure 23.** Modified Normal Mean Bias (left) and correlation (right) of PM2.5 and PM10 based on EEA-NRT-rural (Europe), AirNow (North America) and MEP (East Asia) monitoring stations. The average timeseries for all stations over East Asia (second row) and the MNMB maps for o-suite and e-suite are also shown (third row).





## 4.11 Aerosol coarse

AOD coarse (AODc) of e-suite and o-suite were evaluated against the AERONET SDA version 3 level 1.5 daily data (Fig.
24). Overall, the e-suite AODc performs better than o-suite in terms FGE globally (from 1.09 to 0.77) and for all regions
(except Middle East and Pacific/Australia/New Zealand). The same stands for the MNMB and R (not shown). Since the AE, is
indicative of the aerosol size distribution, its Mean Absolute Bias (MAB) reveals similar results.

Over and around arid areas AODc is represented mainly by aerosol dust. The new dust emission increased Dust Optical Depth
(DOD) over Middle East and increased/decreased the DOD over the northern/southern part of the Sahara (Fig. 20e). Over North
Africa the increasing dust concentration in spring, which increases AODc and decreases AE, is represented better by e-suite
compared to o-suite. Contrary in Middle East e-suite displays too high AODc (too low AE) compare to the observations.
The simulated AODc and AE of e-suite is particularly better than o-suite over AERONET stations that are located over the
westward transport of Saharan dust in N. Atlantic (e.g. Capo Verde), as well as other regions that are affected less from dust
(e.g. southern Europe) (AeroVal, 2023).





**Figure 24.** Fractional gross error (FGE) of AOD coarse based on AERONET SDA (Spectral Deconvolution Algorithm) version 3 level 1.5 daily data (left column) and Ångström Exponent between 440nm and 870nm Mean Absolute Bias (MAB) based on AERONET version 3 level 1.5 daily data (right column). The monthly time series (bold lines) along with the daily time series of observations (thin lines) for N. Africa (second row) and Middle East (third row) are also depicted.



# 5 Conclusions

The upgrade of the ECMWF/CAMS global system to Cy48R1 of 27 June 2023 involved many system changes in the composition modelling, emissions and assimilation which are listed in Section 2.1. The Cy48R1 represent a major upgrade and is the result of two years of model and data assimilation development. The upgrade introduces, among many other changes, a comprehensive stratospheric chemistry scheme with the addition of 63 gas species, contains important emissions updates, implements changes to the modelling of dust aerosol resulting in a redistribution of aerosol particles towards larger sizes and adds an explicit representation of secondary organic aerosol, and revisited isoprene and aromatics chemistry. In Cy48R1 the assimilation of TROPOMI CO and VIIRS AOD is introduced.

The validation results for in total 47 comparisons (measurement datasets) were summarised in the scorecard Table 2, which compares the relative performance of the new Cy48R1 configuration to the previous Cy47R3 system operational until 27 June 2023. The results for the trace gases may be summarised as follows:

- The performance for CO has generally improved against all observations. The comparison against MOPITT is the exception. Previously, the assimilation system was anchored to MOPITT, while since Cy48R1 this is IASI. Furthermore, TROPOMI CO observations are now assimilated in Cy48R1. This anchoring change explains why the comparison with MOPITT is worse in the e-suite compared to the o-suite. However, the independent observations show that these changes are improvements.

- $NO_2$ and $SO_2$ show improved validation results against multiple observations, both at the surface and in columns observed from space. This may be linked to the upgrade of the emissions which are more realistically describing the emission trends in recent years.

- CAMS surface ozone and tropospheric ozone shows similar performance in e-suite and o-suite. However, ozone improvements are observed in Eastern China and the USA at the surface, probably linked to the emission update and changes in the precursors like $NO_2$.

- Stratospheric ozone below the 10 hPa level and total column ozone have improved against observations, especially in the Tropics. This may be linked to the inclusion of a full stratospheric chemistry scheme in Cy48R1. The bug fix in the formulation of the background covariances in Cy47R3 in December 2022 had a positive impact, especially for ozone in the tropics.

- The comparison of a large number of trace species in the stratosphere against ACE-FTS and MLS show realistic concentration profiles. This provides confidence in the implementation of stratospheric chemistry in IFS-COMPO.

- The overestimation of HCHO in the Tropics is more pronounced in the e-suite compare to TROPOMI. This may be related to the updated isoprene chemistry.

- The UV evaluation shows only minor changes.



The results for the aerosols may be summarised as follows:

- The aerosol optical depth evaluation shows improvements for most regions.

- The Ångström exponent, which is a measure representative of the aerosol size distribution, shows some improvement on the global scale but over the Middle East the performance deteriorates showing a too large fraction of coarse particles.

- Particulate matter (PM10 and PM2.5) at the surface is reduced in e-suite, which leads to lower bias in the eastern part of china compared to the o-suite. The e-suite and o-suite exhibits similar performance over North America and Europe.

It is important to note that the e-suite data is available for a period of about 9 months, with more emphasis on the NH Winter season. Therefore some results may not be fully representative for the entire year 2023 or for other years. The assimilation of TROPOMI CO in the e-suite was switched on only at the end of April 2023, and its impact could not be fully evaluated.

In summary, 55% of the evaluation datasets show an improved performance of Cy48R1 compared to the previous operational CAMS system, of which two improvements are indicated as major, 28% of the comparisons are neutral, and 17% indicate a degradation for the Cy48R1 e-suite compared to the Cy47R3 o-suite. This clearly indicates the overall success of the recent upgrade of the CAMS global system to Cy48R1.

The evaluation of the CAMS products with independent observations is continuously developing. Apart from further es-
565 tablishing interfaces with the major observation networks, CAMS is actively acquiring and testing (surface) data from South America, Africa, and Asian countries other than China. The scorecard presented in this paper is providing a qualitative summary of the results. More quantitative scoreboards are being developed for the quarterly o-suite validation reports, see Tsikerdekis et al. (2023).

The next upgrade of CAMS to Cy49R1 is planned for quarter 4 of 2024.

*Code and data availability.* Access to the CAMS daily forecasts and analyses is provided by the CAMS Atmosphere Data Store (ADS) at https://ads.atmosphere.copernicus.eu/, last access: 21 December 2023. The data is available in both GRIB and NetCDF file format. An interactive verification interface for the Cy48R1 e-suite is provided on the AeroVal server of MET-Norway: https://aeroval.met.no/evaluation. php?project=cams2-82&exp_name=IFS-ESUITE-Cy48R1, last access: 21 December 2023. The operational NRT validation server for the CAMS o-suite provides daily-updated comparisons against many of the datasets used in this paper: https://global-evaluation.atmosphere.
copernicus.eu, last access: 21 December 2023. The IFS forecasting and reanalysis system is not for public use as the ECMWF Member States are the proprietary owners. The resulting datasets are however freely available.

*Author contributions.* Henk Eskes and Athanasios Tsikerdekis have initiated the paper and wrote most of the text. Anna Benedictow, Yasmine Bennouna, Lewis Blake, Quentin Errera, Jan Griesfeller, Luka Ilić, John Kapsomenakis, Bavo Langerock, Augustin Mortier, Mikko Pitkänen, Andreas Richter, Anja Schoenhardt, Michael Schulz, Valerie Thouret, Thorsten Warneke and Christos Zerefos have performed
the comparisons and interpretation of the CAMS analysis and forecast results against independent observations. Johannes Flemming, Antje



Inness, Samuel Remy, Vincent Huijnen, Simon Chabrillat, Sebastien Garrigues, Melanie Ades, Zak Kipling, Mihai Alexe, Mark Parrington, Richard Engelen and Vincent-Henri Peuch were involved in the development of the CAMS IFS-COMPO model and data assimilation system. Idir Bouarar, Isabelle Pison were involved in formulating and reviewing the conclusions.

*Competing interests.* At least one of the (co-)authors is a member of the editorial board of Atmospheric Chemistry and Physics.

*Disclaimer.* The study presented in this paper started from the CAMS project document which contains a first evaluation of the Cy48R1 upgrade (Eskes et al., 2023b). In our paper we are presenting new results and figures for an extended period. We note that part of the descriptive text in this paper will resemble text in the CAMS report, available from the CAMS website, written by the same authors.

*Acknowledgements.* We wish to acknowledge the provision of NRT World Meteorological Organisation Global Atmosphere Watch (WMO-GAW) observational data provided to us by Deutscher Wetterdienst (DWD).

We are grateful to the numerous operators of the AERONET network and to the central data processing facility at NASA Goddard Space Flight Center for providing the NRT sun photometer data, especially Ilya Slutker and Brent Holben for sending the data.

The authors thank to all researchers, data providers and collaborators of the World Meteorological Organization's Sand and Dust Storm Warning Advisory and Assessment System (WMO SDS-WAS) for Northern Africa, Middle East and Europe (NAMEE) Regional Node. Also special thank to Canary Government as well as AERONET, MODIS, MSG Eumetsat, EOSDIS World Viewer, the International Cooperative

for Aerosol Prediction (ICAP) principal investigators and scientists for establishing and maintaining data used in the activities of the WMO Barcelona Dust Regional Center (i.e. the WMO SDS-WAS Regional Center for Northern Africa, the Middle East and Europe).

We wish to acknowledge the provision of ozone sonde data by the World Ozone and Ultraviolet Radiation Data Centre established at EC in Toronto (http://woudc.org), by the Data Host Facility of the Network for the Detection of Atmospheric Composition Change established at NOAA (http://ndacc.org), by the Norwegian Institute for Air Research, by the National Aeronautics and Space Administration (NASA)

and the European Validation Data Centre (EVDC, https://evdc.esa.int).

FTIR, MWR, LIDAR, DOBSON, UVVIS DOAS and sonde data used in this publication were obtained from the Network for the Detection of Atmospheric Composition Change (NDACC) and are available through the NDACC website www.ndacc.org.

The authors acknowledge the NOAA Earth System Research Laboratory (ESRL) Global Monitoring Division (GMD) for the provision of ground-based ozone concentrations.

The MOPITT CO data were obtained from the NASA Langley Research Center ASDC. We acknowledge the LATMOS IASI group for providing IASI CO data.

Sentinel-5 Precursor is a European Space Agency (ESA) mission on behalf of the European Commission (EC). The TROPOMI payload is a joint development by ESA and the Netherlands Space Office (NSO). The Sentinel-5 Precursor ground segment development has been funded by ESA and with national contributions from the Netherlands, Germany, and Belgium. This work contains modified Copernicus

Sentinel-5P TROPOMI data (2018–2023). Sentinel-5P lv1 radiances and $NO_2$ operational data were provided by EU Copernicus.



The authors acknowledge Environment and Climate Change Canada for the provision of Alert ozone data and Sara Crepinsek – NOAA for the provision of Tiksi ozone data. Surface ozone data from the Zeppelin Mountain, Svalbard are from www.luftkvalitet.info. Surface ozone data from the Villum Research Station, Station Nord (VRS) were financially supported by "The Danish Environmental Protection Agency" with means from the MIKA/DANCEA funds for Environmental Support to the Arctic Region. The Villum Foundation is acknowledged for

the large grant making it possible to build VRS in North Greenland.

We thank the several institutes, their experts, and partners for providing us with UV index and spectral UV data from their ground based observation stations: Australian Radiation Protection and Nuclear Safety Agency (ARPANSA, Australia), Department for Environment Food & Rural Affaird (DEFRA, UK), Finnish Meteorological Institute (FMI, Finland), Israel Meteorological Service (IMS, Israel), Laboratory of Atmospheric Physics of the Aristotle University of Thessaloniki (Greece), National Institute of Water and Atmospheric Research (NIWA,

New Zealand), Silpakorn University (Thailand)

We acknowledge the National Aeronautics and Space Administration (NASA), USA for providing the OMPS limb sounder data (http://npp.gsfc.nasa.gov/omps.html), the SAGE III-ISS ozone data https://eosweb.larc.nasa.gov/project/sageiii-iss/sageiii-iss_table and the Aura-MLS offline data (http://mls.jpl.nasa.gov/index-eos-mls.php).

We thank the Canadian Space Agency and ACE science team for providing level 2 data retrieved from ACE-FTS on the Canadian satellite

SCISAT-1.

The European Environment Information and Observation Network (EIONET) Air Quality portal provides details relevant for the reporting of air quality information from EU Member States and other EEA member and co-operating countries. This information is submitted according to Directives 2004/107/EC and 2008/50/EC of the European Parliament and of the Council.

We are grateful to the IAGOS operators from the various institutes which are members of IAGOS-AISBL (http://www.iagos.org). The

630 authors also acknowledge the strong support of the European Commission, Airbus, and the airlines (Lufthansa, Air France, Austrian, Air Namibia, Cathay Pacific, Iberia, China Airlines, Hawaiian Airlines, Eurowings Discover and Air Canada) which have carried the IAGOS equipment and undertaken maintenance. IAGOS has been additionally funded by the EU projects IAGOS-DS and IAGOS-ERI. The IAGOS database is supported in France by AERIS (https://www.aeris-data.fr).



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
