# Peer review of "Technical Note: Evaluation of the Copernicus Atmosphere Monitoring Service Cy48R1 upgrade of June 2023"

_EGUsphere, 2023_

## Author Comment (AC1)

**Response to Referee 2:**

General comments:

*The CAMS global forecasts for atmospheric composition are a major effort by the Earth observation component of the European Union's Space programme – Copernicus - to include atmospheric chemistry and aerosols into the global NWP modelling and data assimilation framework, here fully integrated into the IFS modelling system at ECMWF. At the same time, it is a pioneer service to the public, providing various composition data in near-real time alongside with traditional weather prediction. Due to the complex nature of atmospheric chemistry and dynamics and its interplay with the earth surface, great importance must be attached to validation using independent observational data sets. In this manuscript, extensive evaluation has been performed with different types of observations (station networks, sondes, in-situ and remote sensing data). With the update to Cy48R1 several important innovations have been introduced, namely integration of full stratospheric chemistry, emission updates, representation of secondary organic aerosol, and assimilation of new satellite observations for AOD and CO.*

We thank the reviewer for recognising CAMS as a pioneer service, and for stating the importance of the dedicated validation activity.

*Given the broad scope of the research topic, the pioneer nature of the service, and the relevace of the data provided by CAMS, the paper should be published in ACP. However, I suggest to take a few points into consideration before publishing:*

We thank the reviewer for the positive assessment, for a very careful reading of our manuscript, and for all the useful feedback. Incorporating this feedback has improved the paper substantially. We will respond to the considerations below.

*As the authors state in the introduction and in a disclaimer, the manuscript is based on an existing (preliminary) CAMS report (Eskes, 2023). Looking at the results section, more than 20% of the findings in this manuscript are not substantiated with own figures but refer to results presented in Eskes (2023). This may be a fair decision, given the length of the paper. But the reader should be able to easily verify the results. E.g., this can be achieved by adding additional figures to an appendix.*

Following the suggestion of the reviewer we have created an extra document with supplementary material which contains all figures referred to and presented in Eskes (2023). Some of the figures have been re-done to match the looks of the figures in the paper.

*While the general findings are extremely well summarized in the conclusion part, I was not able to fully verify all results, even if there are displayed in the figures. Please refer to my specific*

*comments. A few times, I was also missing the reasoning behind updates or the explanation for differences.*

These specific comments are answered below.

*You also introduce results using the control runs of e-suite and o-suite. While this is nicely introduced in the system overview, later on it often remains unclear why these simulations are shown. Please revise all result sections, keep and explain results which can be achieved with the help of the control runs only and remove those results elsewhere. Ensure that the figures contain relevant information only, but try to enrich the statistical information with some overall values (see specific comments).*

Statistical information (mean and standard deviation) is now added to close to half of the figures. In general we think it is useful to show also the results for the control runs, which in most cases brings interesting additional information, distinguishing model changes from assimilation impacts. The discussions on the results have been reviewed, see responses to the specific comments below.

*Specific comments:*

*Line 26: Add MACC reference, e.g., Marécal et al. (2015).*

We added "CAMS is furthermore producing daily forecasts and analyses of air quality in Europe based on an ensemble of air quality models (Marecal et al., 2015)." Note that this is an old paper, but still the main overview paper for the regional system. An update of this paper is in preparation.

*Line 32: As reference, you could point to:  https://www.ecmwf.int/en/forecasts/documentation-and-support/changes-ecmwf-model*

A link to the ECMWF online documentation was added. (ECMWF: Changes to the forecasting system, https://confluence.ecmwf.int/display/FCST/Changes+to+the+forecasting+system, 2024h)

*Line 34: To be more precise, change to "The assimilation of satellite data for atmospheric composition …"*

Suggestion included in revised manuscript

*Line 38: Remove "high-resolution".*

We left the text as it is. CAMS is producing the GHG forecasts at two resolutions, one at high resolution with a grid spacing of about 9km (TCo1279) with 137 layers.

*Line 43: TROPOMI and VIIRS instruments need to be introduced, AOD abbreviation needs to be explained.*

Done

*Lines 47/48: You should add here the work by Katragkou et al. (2015) on evaluation of near-surface ozone over Europe.*

Thanks for pointing out this omission. Reference to Katragkou is now included.

*Lines 49/50: These organisations need to be introduced. You can do that by refering to Table 1.*

The full name of the organisations has been added.

*Line 50: "real-time" should be "near-real time".*

Done.

*Line 51: Please change reference to ECMWF (2023d) to account for the series character.*

Done.

*Line 52: The validation report on the CAMS GHG reanalysis by Ramonet et al. (2021) has not been updated so far. Moreover, this reanalysis seems to have being stopped by the end of 2020. Please clarify or omit.*

We kept the reference (it is the latest report) and added the following text: "Note that the production of the GHG reanalysis was interrupted (Agusti-Panareda et al., 2023) for a few years due to a degraded quality of the input satellite data."

*Line 62: The term "compo-suite" is not used further and can be omitted.*

Done.

*Lines 100-103: This important introduction into the nature of the control runs should be placed directly after its first mention (after line 92).*

Done.

*Lines 104-106: The relation of IFS-COMPO to IFS-GHG is not well introduced with the description of CAMS services. You can add one or two sentences in the introduction and skip this paragraph here.*

Good suggestion. The paragraph was removed and updated info was added to the introduction:

"The CAMS pre-operational analyses and forecasts of CO2 and CH4 use an independent setup of the IFS. The upgrade of the greenhouse gas system to Cy48R1 took place on 27 February 2024, and a separate upgrade verification report was written (Langerock et al., 2024). The greenhouse gas products will not be discussed in this paper."

*Lines 108-109: The cycles have already been explained in the introduction. Skip the sentence here.*

Done.

*Line 118: BASCOE abbreviation needs to be explained here (done in line 124).*

Done.

*Line 125: 64 species are mentioned here to be used in BASCOE, in line 118/119 you talk about 123 active tracers to be included. Following Errera et al. (2019), 58 species are included. As the integration of a full stratospheric chemistry scheme is a major achievement of the latest CAMS update, please be more precise here.*

The 123 tracers is the combination of all tracers that are chemically active in either the troposphere, the stratosphere, or both. The 64 tracers listed active in stratospheric chemistry include HCN and CH3CN, as well as tracers to represent sulphur chemistry, which are additions specific to CAMS, on top of the default BASCOE chemistry described in Errera et al (2019). This explains the larger number of tracers involved compared to the mechanism described in Errera et al., (2019). The tropospheric chemistry involves 71 tracers, which is an expansion compared to the previous cycle by 16 tracers, associated to the revision of isoprene chemistry, the addition of new aromatics tracers and the inclusion of SOA precursor gas tracers.

In the manuscript we added the following text:
"Note that the 64 tracers listed as active in stratospheric chemistry include HCN and CH3CN, as well as the tracers to represent sulphur chemistry, which are additions specific to CAMS, on top of the default BASCOE chemistry with 58 species described in Errera et al (2019)."

*Lines 170-107: MODIS, S-NPP, and NOAA20 need to be introduced.*

Done

*Line 174: AERONET needs to be introduced (-> Table 1).*

Done

*Line 194: You should add here that diurnal cycle profiles are partly applied to the emissions, depending on sector and species.*

You are correct, but note that in fact, this aspect was already in the text, see line 200. A reference to Guevara was added. Line 200 now reads: "In Cy48r1 a sector-specific treatment for any of the emissions is introduced, allowing sector-specific diurnal cycle profiles and injection heights, see section 3.1.1 in ECMWF (2023) and Guevara et al. (2021) for more details.

*Lines 194-198: Are these emission datasets published and/or citeable? In the IFS documentation for Cy38R1 the following references are given: Granier et al. (2022) and Denier van der Gon (2021), which both refer to unspecified CAMS reports.*

For the CAMS-GLOB-ANT v5.3 the paper of Soulie et al. (ESSD 2023) is now accepted and will appear in final form soon (see https://essd.copernicus.org/preprints/essd-2023-306/). For CAMS-GLOB-BIO v3.1 there is the following reference: https://essd.copernicus.org/articles/14/251/2022/. Both references have been added to the paper.

*Lines 203-206: Latest validation results can be found in ECMWF(2023), while Benedictow et al. (2023) is only one report out of a series. The latter describes validation of Cy47R3. Please change accordingly.*

Indeed this is no longer the latest report. Reference has been updated, and a reference to the page with all reports has been added.

*Line 220: How do you define „main trace gases and aerosol"? I'd argue that this selection is at least partly due to observational constraints.*

Agree. The sentence has been reformulated to "This is presented for individual trace gases and aerosol properties, for the available observational data sets and for regions of interest."

*Line 236: Don't mention the control runs here as they are not shown in Fig.2.*

The new figure 2 now included the control run results.

*Line 251: What does manual validation by the PI imply for IAGOS L1 data?*

The IAGOS NRT data are only available for operational users such as the weather and air quality services (e.g. CAMS). These data are further inspected and validated by the instrument PI using a semi-automatic approach. The data then become available as preliminary data Level 1 (L1) after a time-delay of a few days. This first stage in the QA/QC procedure is fully described in Nédélec et al. (2015) for ozone and CO. At the end of its operational period, the instrument is removed from the aircraft and post-flight calibrated by the PI in the laboratory to produce Level 2 (L2) data. Due to L2 availability delay, only L1 can be used for the validation of CAMS operational suites, while L2 data are usually ready for the evaluation of the reanalyses. The concept and description of these data levels as used in the IAGOS database are detailed in Petzold et al. (2015).

In the paper we added the text:

"The CAMS daily forecast and analysis products are evaluated on a regular 3-monthly basis. In practice this implies that only datasets can be used that are available within one month after real time. With several networks, such as NDACC, IAGOS or EEA surface observations, special arrangements (contracts) have been made such that near-real time unvalidated data can be used. For instance, the IAGOS NRT data are only available for operational users such as the weather and air quality services (e.g. CAMS). These data are inspected and validated by the instrument PI using a semi-automatic approach, and become available as preliminary Level 1 (L1) data with a time-delay of a few days, as described in Nedelec et al. (2015) for ozone and CO, and in Petzold et al. (2015). "

Nédélec M, Halson S, Abaidia AE, Ahmaidi S, Dupont G. Stress, Sleep and Recovery in Elite Soccer: A Critical Review of the Literature. Sports Med. 2015 Oct;45(10):1387-400. doi: 10.1007/s40279-015-0358-z. PMID: 26206724.

Petzold, A., Thouret, V., Gerbig, C., Zahn, A., Brenninkmeijer, C. A. M., … Gallagher, M. (2015). Global-scale atmosphere monitoring by in-service aircraft – current achievements and future prospects of the European Research Infrastructure IAGOS. Tellus B: Chemical and Physical Meteorology, 67(1). https://doi.org/10.3402/tellusb.v67.28452

*Line 253: For comparison with the figure, better stick to altitude here (e.g., 1.5 to 8 km).*

The height range was added.

*Lines 288-289: I guess you mean "The e-suite shows similar correlations as the o-suite with values exceeding 0.6 for most stations". I cannot confirm this statement. Correlations are varying from 0 to ~0.8. Please revise or clarify.*

The text was adjusted to "Overall correlation range between 0.3 to 0.8 for most stations with a mean of 0.53 for the e-suite. The e-suite shows similar correlations as the o-suite. In the densely populated north-eastern part of China correlations are exceeding 0.6 over most stations."

*Lines 300-301: I can't see any obvious differences in bias between e-suite and o-suite in Fig.8.*

There is a clear signal and increase of the MNMB, which can also be observed in the figure, although we agree that the results look similar at first glance. The improvement is more clearly seen in the FGE comparison in the bottom row of Fig. 8.

*Lines 304-310: This refers to Fig. 8, bottom row (?). I can't see the differences described here in Fig.8. Please revise the whole section.*

Note that the figure we referred to in the old text is now included as figure S9 in the new supplement. This figure clearly shows the significant and noteworthy improvement in MNMB

and FGE in Frankfurt, consistent with the original text. Therefore we did not make any changes in this paragraph.

*Lines 319-321: Please mention already at the beginning of the sentence that you refer to the relative bias in December 2022 (Fig.9, third row). The bias south of 60°S is negative, not positive.*

Done. The sentence now reads: "The e-suite relative bias for December 2022 with reference to IASI stays mostly within 20%, with wide areas below 5% and some negative bias above the Southern Africa biomass burning area (up to 20%), and a positive bias around 60°S, and a negative bias over Antarctica."

*Lines 323-324: There are larger negative biases for e-suite in March and June 2023, particularly over the Arctic Sea (March) and North America (June), which are neither described nor explained.*

We added a paragraph on the Canadian fires in the manuscript:
"In 2023 Canada was suffering from a very extreme fire season, which started already in May, with intense fires in June and continuing during the Summer. The e-suite is showing pronounced enhancements in CO, linked to the large GFAS fire emissions over Canada inserted in the model. As shown in the figure, the amount of CO produced in June is underestimated by up to 20\% compared to IASI, and the largest relative error in the global map is located over Canada in June. However, the e-suite compares well with MOPITT over Canada, demonstrating the differences between these two satellite products. Negative CO biases over Canada compared to IASI (but positive compared to MOPITT) are also found in July and August, as shown in the JJA validation report available at  ECMWF (2024d)."

We also added text to discuss the Arctic Sea:
"Over the Arctic Sea we observe a large region with negative biases in March compared to IAS. Note that IASI data has less sensitivity in spring as compared to summer above the Arctic due to lower thermal contrast, which may explain part of the differences. The MOPITT results seem to indicate a similar underestimate, so a negative bias in the analysis is likely."

*Line 331-343: CO partial columns are not shown in Fig.10. Most of the section describe these partial columns. It remains unclear, if Lines 340-343  refer to Fig.10.*

We added tables S1 and S2 to the supplement, containing the scores for the tropospheric and stratospheric partial columns for 13 NDACC stations.

*Line 419: I guess that you mean here the introduction of full stratospheric chemistry. Please confirm.*

Yes. This has been made more explicit in the revised text.

*Line 464: Changes in Nitrates AOD between e-suite and o-suite over southeast Asia and Sahel are in the same order than the absolute values for e-suite. Please revise.*

Revised to: "Over southeast Asia and Sahel nitrates optical depth decreases by about 50% in the e-suite compared to the o-suite."

*Line 466: Add for AE: "… , a parameter which is indicative of the aerosol size distribution, …" and remove this part in line 480.*

Following the suggestion, we moved the sentence to line 466 and removed it from 480.

*Lines 489-490: Network abbreviations have been explained before (or should have been).*

All abbreviations have been checked. They are now detailed on their first occurrence in the text.

*Line 509: "SDA" needs to be explained here (done in Fig.24).*

Abbreviation full name has been added here. "Spectral De-convolution Algorithm (SDA)".

*Lines 514-516: Both, e-suite and o-suite perform moderately over North Africa. I would not set any preference from Fig. 24 (middle row).*

Indeed both runs have moderate performance over North Africa. However, especially for the AE the e-suite is less biased (high) compared to the o-suite. We have altered the sentence to "Over North Africa the increasing dust concentration in spring, which increases AODc and decreases AE, is represented slightly better by e-suite compared to o-suite.".

*Table 1: The link to https://www.eumetsat.int/iasi reveals "access denied".*

We now refer to the ESA webpage in IASI,
https://www.esa.int/Applications/Observing_the_Earth/Meteorological_missions/MetOp/About_IASI

*Table 2: What objective criteria were used to produce the relative scores for the evaluation? How is "small" and "significant" defined?*

Table 2 is providing the summary of all comparisons conducted. We added a paragraph in the conclusion section to comment on the criteria for arriving at the relative score:
"The judgements presented in this summary table are qualitative, based on expert judgement and the investigation of maps and scores. The presentation of one quantitative score and significance assessment performed on all datasets, is in practice not feasible or meaningful. Each entry in the table has its own story, which is detailed in the results sections above. The datasets used for the assessment are very heterogeneous. Some datasets are very sparse (e.g. IAGOS where an extended data record is only available over Frankfurt) or cover only a limited

region (e.g. surface concentrations from air-quality networks), while other networks are dense, such as AERONET. Satellite data is available globally, but retrieval products for separate instruments (or the same instrument) often show significant differences and biases and uncertainties are not easy to assess. There are strong regional and seasonal differences in the scores reflecting differences in sources, processes, or aspects like aerosol composition."

*Figures 1,2,3,7,11,13,21,22,23: Always specify clearly the two datasets where your statistical evaluation is based on. E.g., "e-suite – observations" in Fig.1, top row.*

The captions of these 9 figures have been adjusted accordingly, mentioning the bias refers to CAMS-observation.

*Figures 1,2,3,7,11,13,21,22,23: Mean value over all stations of the statistical measure shall be given as numbers in each panel, ideally together with standard deviation.*

All these figures have been updated, and the mean and the standard deviation have been added inside these figures.

*Figures 1,2,3,7,11,13,21,22,23: Background of the upper left corner in all panels is either light gray or light blue. Panel enumeration would be better visible with black colour letters.*

We changed the font colour to black for all of these figures, as requested.

*Figure 10: "GB" (Global Bias?) needs to be introduced or omitted. "GB_origpressure" should be "pressure"?*

The figure has been updated. The y-axis label now says "pressure (hPa)". "GB_origpressure" means that the pressure grid of the ground based measurements was used as vertical coordinate.

*Figure 17: Specify blue lines and black dots in the caption.*

These are now specified in the caption of Figure 17.

*Figure 18: I assume that the stratospheric NO2 columns  shown in the top row are the same than used for the differences between e-suite and TROPOMI. For the bottom row you mention the SREAM-B algorithm explicitly. Is there any reason for that?*

Figure 18 is replaced and no longer mentions STREAM-B.

*Figure 19: Better use "o-suite" in x-axis labeling.*

The figure has been updated. The x-axis label is now "o-suite".

*Figure 20: DU, SS, OM, BC, SU, NI, AM need to be introduced.*

These are now explained in the caption of Fig. 20.

*Figure 23: Give MNMB and Correlation as titles to the top row. Here, it would be sufficient to show PM2.5 only.*

Labels for MNMB and R have been added above the colour legend of each figure. PM10 and PM2.5 show substantially different statistics, and therefore we prefer to keep the PM10 results in the table.

*Technical comments:*

*Line 144: Replace "… that used …" by "… that is used …".*
*Line 159: Put "since cycle 46R1" in brackets.*
*Line 165: Add the year 2023 to the date.*
*Line 177: Replace "VarBC involves introducing …" by "VarBC introduces …".*
*Line 185: "and" -> "an"*
*Line 214: "show" -> "shown"*
*Line 231: "smaal" -> "small"*
*Line 471: Replace "except dust" by "except for dust".*
*Line 486: Put "AeroVal (2023)" in brackets.*
*Line 511: Remove , after "AE".*
*Line 519: Replace "Capo Verde" by "Cabo Verde".*
*Figure 5: Caption: "covariancein" -> "covariance in"*

All these corrections have been implemented in the revised manuscript. Thanks for a careful reading!

**References:**

*Benedictow, A., Arola, A., Bennouna, Y., Bouarar, I., Cuevas, E., Errera, Q., Eskes, H., Griesfeller, J., Basart, S., Kapsomenakis, J., Lange[1]rock, B., Mortier, A., Pison, I., Pitkänen, M., Ramonet, M., Richter, A., Schoenhardt, A., Schulz, M., Tarniewicz, J., Thouret, V., Tsik[1]erdekis, A., Warneke, T., and Zerefos, C.: Validation report of the CAMS near-real-time global atmospheric composition service, Period December – February 2023, https://doi.org/10.24380/i31d-5i54, 2023.*

This is no longer the latest report. The reference has been replaced by "Warneke et al., 2024".

*ECMWF: CAMS global validation services, https://atmosphere.copernicus.eu/global-services, 2023.*

Note that all references to web pages on the ECMWF/CAMS website have been checked, and the year is now "2024".

*Errera, Q., Chabrillat, S., Christophe, Y., Debosscher, J., Hubert, D., Lahoz, W., Santee, M. L., Shiotani, M., Skachko, S., von Clarmann, T., and Walker, K.: Technical note: Reanalysis of Aura MLS chemical observations, Atmos. Chem. Phys., 19, 13647–13679, https://doi.org/10.5194/acp-19-13647-2019, 2019.*

*Eskes, H., Tsikerdekis, A., Benedictow, A., Bennouna, Y., Blake, L., Bouarar, I., Errera, Q., Griesfeller, J., Ilic, L., Kapsomenakis, J., Langerock, B., Mortier, A., Pison, I., Pitkänen, M., Richter, A., Schönhardt, A., Schulz, M., Thouret, V., Warneke, T., and Zerefos, C.: Upgrade verification note for the CAMS near-real time global atmospheric composition service: Evaluation of the e-suite for the CAMS CY48R1 upgrade of 27 June 2023, https://doi.org/10.24380/rzg1-8f3l, 2023.*

*Katragkou, E., Zanis, P., Tsikerdekis, A., Kapsomenakis, J., Melas, D., Eskes, H., Flemming, J., Huijnen, V., Inness, A., Schultz, M. G., Stein, O., and Zerefos, C. S.: Evaluation of near-surface ozone over Europe from the MACC reanalysis, Geosci. Model Dev., 8, 2299–2314, https://doi.org/10.5194/gmd-8-2299-2015, 2015.*

This paper has been added

*Marécal, V., Peuch, V.-H., Andersson, C., Andersson, S., Arteta, J., Beekmann, M., Benedictow, A., Bergström, R., Bessagnet, B., Cansado, A., Chéroux, F., Colette, A., Coman, A., Curier, R. L., Denier van der Gon, H. A. C., Drouin, A., Elbern, H., Emili, E., Engelen, R. J., Eskes, H. J., Foret, G., Friese, E., Gauss, M., Giannaros, C., Guth, J., Joly, M., Jaumouillé, E., Josse, B., Kadygrov, N., Kaiser, J. W., Krajsek, K., Kuenen, J., Kumar, U., Liora, N., Lopez, E., Malherbe, L., Martinez, I., Melas, D., Meleux, F., Menut, L., Moinat, P., Morales, T., Parmentier, J., Piacentini, A., Plu, M., Poupkou, A., Queguiner, S., Robertson, L., Rouïl, L., Schaap, M., Segers, A., Sofiev, M., Tarasson, L., Thomas, M., Timmermans, R., Valdebenito, Á., van Velthoven, P., van Versendaal, R., Vira, J., and Ung, A.: A regional air quality forecasting system over Europe: the MACC-II daily ensemble production, Geosci. Model Dev., 8, 2777–2813, https://doi.org/10.5194/gmd-8-2777-2015, 2015.*

This paper has been added:

*Ramonet, M., Langerock, B., Warneke, T., and Eskes, H.: Validation report of the CAMS greenhouse gas global reanalysis, years 2003-2020,*

This is the latest report. In the revised text we explain that the production was discontinued for two years, and that an update of this validation report will become available soon.

---

## Author Comment (AC2)

**Response to Referee 1:**

*This manuscript documented the performance of the CAMS Cy48R1 in comparison with CAMS Cy47R3. This is an important work, and the paper is well structured. I only have a few minor comments.*

We thank the referee for the recognition of our work, and for the compliment on the structure of the paper.

*Out of curiosity, what's the meaning of Cy47R3, Cy48R1?*

The cycles in CAMS follow the cycles of the ECMWF Integrated Forecasting System IFS. The cycle numbers are basically version numbers, where the first number is the major (yearly) upgrade number, and R1, R2 etc refer to typically smaller updates during the year. ECMWF is approaching its 50th birthday, and the cycles also approach 50.

The upgrades are discussed in lines 55-60 of the introduction. We added: "Detailed information about the ECMWF IFS upgrades can be found on ECMWF (2024h)".

*Line 49: add full name of NDACC, WMO-GAW, AERONET, IAGOS, ICOS, IASOA?*

Full names have been added in the revised manuscript. Same for NASA, ESA, EUMETSAT.

*As noted by the authors, summer validation over the Northern Hemisphere (winter for the Southern Hemisphere) is missing. Please add some more discussion on this. Is the performance in the missing months expected to be similar to other months?*

Unfortunately only 9 months of data are available to us. Indeed, for atmospheric composition we may expect seasonally-dependent validation results. We have added the following text: "The choice to generate 9 months of e-suite data is made based on practical timing and computational resources considerations. The length of the e-suite run has been discussed in the CAMS team, and for the next upgrade to Cy49R1, planned for the end of 2024, the e-suite will hopefully cover a full year."

*Do you have any assimilation plan for the geostationary satellites?*

Yes. The following has been added:
"CAMS is continuously extending its activity by testing and using new emerging datasets such as trace gas retrievals from the geostationary Geostationary Environmental Monitoring Spectrometer (GEMS) and Tropospheric Emissions: Monitoring of Pollution (TEMPO), improved retrievals from past and present missions, and preparing for future missions such as Sentinels 4 and 5."

*Line 230: "smaal" typo?*

Corrected

*It will be helpful to have a table of all assimilated satellite products in CAMS in the previous and current version, and their assimilated period.*

Instead of a table, we have added Figure S2 in the supplement. It shows which satellite data products are assimilated and for which period. The latest additions are VIIRS AOD and TROPOMI CO, as discussed in the paper.

*Figure 2: Why e-control – o-control is not shown over Asia like Figures 1 & 3?*

The e-control – o-control is now added to figures 1 and 3.

*Line 259-261: "At most airports worldwide the bias in the lower troposphere (pressure > 850 hPa) is slightly larger for the e-suite than for the o-suite, and in particular over airports located in Western Africa and Eastern Asia (not shown). Conversely in the free troposphere, the bias is smaller in the e-suite that in the o-suite for most visited airports (Eskes et al., 2023b)." Why is that the case? Overall, the paper focused on evaluating the performance of the e- suite relative to o- suite but often does not explain the differences.*

The text has been reformulated and now reads:
"In the free troposphere, the bias is about the same between the e-suite than in the o-suite for most visited airports for the analysis or the 1-day forecast (Figure S5). For the control run we observe larger differences, and the o-control shows mainly positive biases, the e-control mainly negative biases compared to IAGOS, see also Fig. 4. These differences are reduced and results are improved by the data assimilation in the free troposphere."

In general, ozone is influenced by many aspects including emissions, implementation of the chemistry, changes in precursor gases etc. Therefore it is very difficult to say what is causing the observed ozone changes, especially because many aspects were changed in the Cy48R1 upgrade. Interestingly the data assimilation brings the ozone results of the e-suite and o-suite closer to each other, improving both.

*Change "ozone sonde" to "ozonesonde"?*

In the original paper we used a mix of both. Now all occurrences of "ozone sonde" have been replaced by "ozonesonde", as suggested by the referee.

*Line 556: change "china" to "China"*

Done

*I like it that the author mentioned in the conclusion "CAMS is actively acquiring and testing (surface) data from South America, Africa, and Asian countries other than China." I hope in the future more data from the Global South will be added in the evaluation.*

This is certainly the plan. We are currently assessing the quality of a number of candidate measurement datasets.

*The manuscript is very long with many details. While this might be the nature of such type of papers (Technical Note), it is not easy to follow the whole text for general readers. The abstract and conclusion are helpful because they listed main points.*

We have been struggling to find the right balance by not providing too much, or too little results. The comparison with so many independent measurement datasets for a number of species and aerosol properties generated many results. At the same time we aimed at providing a full overview of the activity. We have carefully selected only those figures which document the main conclusions. Note that referee 2 asked us to add those figures from the report (Eskes, 2023) that are referred to in the paper. In response we have created a supplementary material document which contains those figures.

We are glad that the reviewer appreciates the summaries in the abstract and conclusion, which we tried to keep short and to the point.